# Mem-$\alpha$: Learning Memory Construction via Reinforcement Learning

## Abstract

Large language model (LLM) agents are constrained by limited context windows, necessitating external memory systems for long-term information understanding. Current memory-augmented agents typically depend on pre-defined instructions and tools for memory updates. However, language models may lack the ability to determine which information to store, how to structure it, and when to update it—especially as memory systems become more complex. This results in suboptimal memory construction and information loss. To this end, we propose Mem-$\alpha$, a reinforcement learning framework that trains agents to effectively manage complex memory systems through interaction and feedback. We also construct a specialized training dataset spanning diverse multi-turn interaction patterns paired with comprehensive evaluation questions designed to teach effective memory management. During training, agents process sequential information chunks, learn to extract, store, and update the memory system. The reward signal derives from downstream question-answering accuracy over the full interaction history, directly optimizing for memory construction. To illustrate the effectiveness of our training framework, we design a memory architecture comprising core, episodic, and semantic components, equipped with multiple tools for memory operations. Empirical evaluation demonstrates that Mem-$\alpha$ achieves significant improvements over existing memory-augmented agent baselines. Despite being trained exclusively on instances with a maximum length of 30k tokens, our agents exhibit remarkable generalization to sequences exceeding 400k tokens—over 13× the training length, highlighting the robustness of reinforcement learning for memory management. Code and data will be released upon publication.

## 1 Introduction

Large language model (LLM) agents are fundamentally constrained by limited context windows when processing long information streams, leading to the development of memory-augmented agents (Wang et al., 2025/02; Fang et al., 2025). These agents are equipped with persistent, updatable memory systems that actively stores long-term information and manage the context seen by the language model (Packer et al., 2023; Lin et al., 2025; Cai et al., 2025). Most existing memory systems rely entirely on pre-defined instructions and fixed tool sets without any training to optimize memory construction, such as Mem0 (Chhikara et al., 2025), MemGPT (Packer et al., 2023), and MIRIX (Wang & Chen, 2025). These memory systems provide agents with various memory update tools—ranging from simple fact extraction to complex multi-component memory architectures—but expect models to utilize these tools effectively out-of-the-box. However, models lack the inherent ability to determine what to store, how to structure, and when to update different memory components. Although complicated system prompts can partially mitigate this issue, manual adjustment of system prompts is challenging to address all scenarios. For small language models with weak instruction-following abilities, complicated instructions may even confuse the model (Wen et al., 2024; Wang et al., 2025b).

To address this challenge, we turn to reinforcement learning (RL) as a principled approach for training agents to learn effective memory management strategies. Unlike supervised fine-tuning, which requires ground-truth memory construction traces, RL enables agents to discover optimal memory strategies through trial and error. This approach is necessary across all model scales: even state-of-the-art models like GPT-4o struggle with proper tool selection for memory updates (Wang & Chen,

"[Dialogue at timestamp 2023/03/08 (Wed) 01:55]
<User>I'm looking to get some advice on condo living. Do you have any tips on how to minimize noise pollution in a condo?
<Assistant>I'm glad you asked! … Good luck with your condo search, and I hope you find the perfect one soon!" (1923 tokens)

**Qwen3-4B Before RL**

Calling tools:

memory_update       memory_insert

Core Memory     Episodic Memory     Semantic Memory

❌

Reason: No Update on Core Memory, Missing important information.

At timestamp 2023/03/08 (Wed) 01:55, User asked for … ✅

User is seeking advice on noise pollution … ❌

Reason: Only one entry saved, leading to significant information loss.

**Qwen3-4B w/ MemCraft**

Calling tools:

memory_update       memory_insert

Core Memory     Episodic Memory     Semantic Memory

User cares about noise pollution. User is recently looking for advice on condo living. ✅

At timestamp 2023/03/08 (Wed) 01:55, User asked for … ✅

User is seeking advice on noise pollution … ✅

Tips: Noise Cancellation… ✅

Tips: Neighbor Evaluation… ✅

Figure 1: **Reinforcement learning teaches agents to select appropriate memory tools and types.** Before training (left), agents struggle with tool selection when given new information. After RL training (right), agents learn effective memory management policies.

2025), while smaller models become completely overwhelmed by complex tool sets (Wang & Chen, 2025; Wang et al., 2025b). Since we cannot obtain reliable supervision signals from any existing model, we instead directly optimize for downstream task performance—using question-answering accuracy and memory quality metrics as reward signals. Through RL, language models learn to navigate complex memory systems effectively, discovering strategies that optimize memory construction without relying on potentially suboptimal predefined behaviors. Existing works including MEM1 (Zhou et al., 2025), MemAgent (Yu et al., 2025) and Memory-R1 (Yan et al., 2025) are the first works exploring this direction. However, they employ relatively simple memory structures (e.g., memory rewriting or maintaining a list of facts) that are insufficient for handling complex data such as long narratives, procedural rules, evolving knowledge, or even multi-modal information.

To this end, we propose Mem-$\alpha$, a reinforcement learning framework that trains agents to effectively manage complex memory systems through interaction and feedback. Unlike existing approaches that either provide sophisticated tools without teaching models how to use them, or train models on simplistic memory operations, Mem-$\alpha$ enables agents to learn memory construction strategies for complex, multi-component memory architectures (as shown in Figure 1). Our approach addresses three key challenges in memory-augmented agent training. First, we formulate the process of memory construction as a sequential decision-making problem where agents process information chunks, decide which memory operations to perform, and receive multiple rewards based on downstream question-answering accuracy over the full interaction history. This direct optimization for end-task performance naturally teaches agents to save the most important information and organize the existing memory effectively. Second, we construct a specialized training dataset spanning diverse multi-turn interaction patterns, including conversations, document sharing, pattern recognition, and storytelling, paired with comprehensive evaluation questions that require comprehensive memory to answer correctly. This design exposes agents to various scenarios of memory management during training. Lastly, we adopt a comprehensive memory architecture comprising core, episodic, and semantic components, each equipped with specialized tools for memory operations, providing sufficient expressiveness to handle diverse information types while remaining learnable through reinforcement.

Empirical evaluation demonstrates that Mem-$\alpha$ achieves significant improvements over existing memory-augmented agent baselines across diverse benchmarks. Most remarkably, despite being trained exclusively on instances with a maximum length of 30k tokens, our agents exhibit robust generalization to sequences exceeding 400k tokens, over 13× the training length. This exceptional length generalization suggests that reinforcement learning enables agents to learn fundamental memory management principles rather than merely memorizing specific patterns, highlighting the potential of learning-based approaches for long-context retention.

## 2 RELATED WORK

**Latent-Space Memory** These methods encode new information directly into a model's internal components—such as hidden states (Wang et al., 2024; 2025a; Bulatov et al., 2022; He et al., 2024), key-value caches (Qian et al., 2025; Li et al., 2024; Zhang et al., 2023b; Zhong et al., 2023), soft prompts (Burtsev & Sapunov, 2020; Ge et al., 2023), model parameters (Behrouz et al., 2024; Berges et al., 2024; Wang et al.; Wei et al., 2025), or learnable external matrices (Das et al., 2024). The main advantage is efficient compression: for instance, SELF-PARAM (Wang et al.) can memorize hundreds of contexts without external storage. However, these approaches face two key limitations. First, their memory capacity remains bounded—M+ (Wang et al., 2025a) achieves retention of approximately 160k tokens, which falls short of state-of-the-art memory agents like MIRIX (Wang & Chen, 2025). Second, they require direct access to model internals, making them incompatible with proprietary systems (e.g., GPT-4/5). Since open-weight alternatives typically underperform leading proprietary models, these constraints limit practical deployment.

**LLM Agents with External Memory** An alternative approach equips language models with external memory systems built on databases or vector stores (Zhang et al., 2025a), as demonstrated by MemGAS (Xu et al., 2025a), SCM (Wang et al., 2023), A-MEM (Xu et al., 2025b), MemTree (Rezazadeh et al., 2024) MemGPT (Packer et al., 2023), Mem0 (Chhikara et al., 2025), Zep (Rasmussen et al., 2025), Nemori (Nan et al., 2025), EgoMem (Yao et al., 2025), MIRIX (Wang & Chen, 2025), Memobase[1], MemoChat (Lu et al., 2023) and similar frameworks. These architectures offer two key advantages: they work seamlessly with proprietary frontier models (e.g., GPT-4/5, Claude family) and can efficiently organize, retrieve, and update large amounts of information through well-designed schemas and controllers. However, their effectiveness depends heavily on the base model's ability to follow instructions and use tools (function-calling)—capabilities that smaller, more cost-effective models often lack. Meanwhile, when the system becomes complex, even proprietary models may not update the memory systems well (Wang & Chen, 2025). This limitation motivates approaches that explicitly *train* models to manage memory rather than relying purely on prompting.

**Learning Memory Construction with Reinforcement Learning** Recent work explores training language models to construct memory using reinforcement learning, though results remain preliminary. Early efforts such as MEM1 (Zhou et al., 2025) and MemAgent (Yu et al., 2025) train models to update simple, text-only memories. Memory-R1 (Yan et al., 2025), Learn-to-Memorize (Zhang et al., 2025b) and REMEMBER (Zhang et al., 2023a) introduce a slightly richer memory representation and a simplified tool-calling interface, but focuses on LoCoMo (Maharana et al., 2024) settings with relatively short maximum context (less than ∼26k tokens) and train on subsets of the same distribution, which makes the task comparatively easier. In this paper, we develop an RL framework that trains a model to operate a substantially more capable memory system and demonstrate significant improvements across multiple dimensions of memory quality and efficiency.

## 3 METHOD

### 3.1 REINFORCEMENT LEARNING FRAMEWORK

We formulate memory construction as a reinforcement learning problem where the agent learns to optimize memory building policies. The quality of the constructed memory is evaluated through a separate question-answering process using retrieval-augmented generation (RAG). The complete training framework is shown in Figure 2.

#### 3.1.1 TASK SETUP

We consider a memory construction task where an agent processes a sequence of conversations $\mathcal{C} = \{c_1, \ldots, c_n\}$ between a user and an assistant. These conversations span diverse formats, including casual discussions, storytelling, book sharing, and classification examples. At step $t \in \{1, \ldots, n\}$ the agent observes $c_t$ and the current memory $\mathcal{M}_{t-1}$ (here $\mathcal{M}$ is the memory and $\mathcal{M}_0$ is initialized as an empty memory) and may issue a *sequence* of write operations before advancing to the next

---

[1]https://github.com/memodb-io/memobase

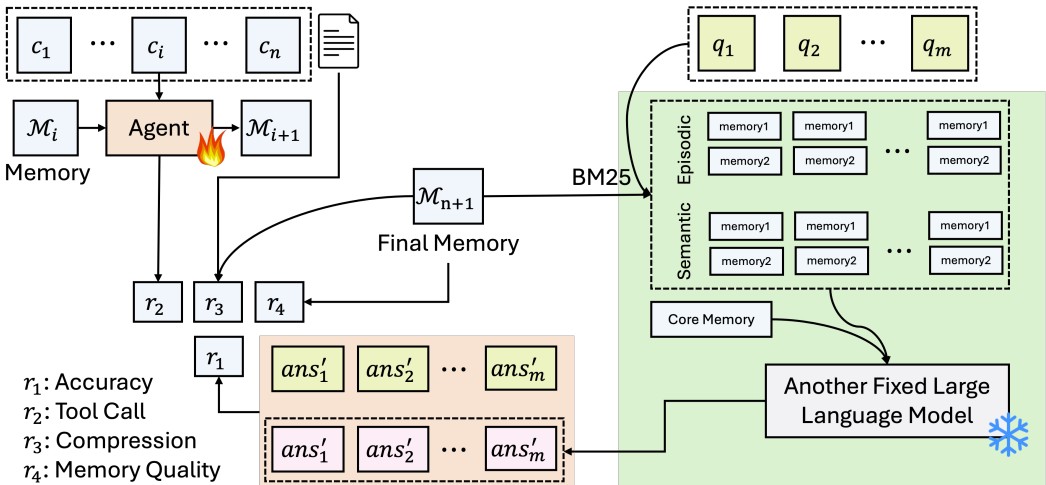

Figure 2: **Training Framework of Mem-$\alpha$.**

chunk. Formally, the action at step $t$ is

$$a_t = \left(a_t^{(1)}, \ldots, a_t^{(K_t)}\right)$$

where each $a_t^{(k)} \in \mathcal{A}_{\text{write}} = \{\texttt{memory\_insert}, \texttt{memory\_update}, \texttt{memory\_delete}\}$ is a structured function call with arguments (e.g., record id, memory type, string content), and $K_t$ is the number of operations in this action. Then we apply these function calls on $\mathcal{M}_{t-1}$:

$$\mathcal{M}_{t-1}^{(0)} = \mathcal{M}_{t-1}, \qquad \mathcal{M}_{t-1}^{(k)} = T\left(\mathcal{M}_{t-1}^{(k-1)}, a_{t-1}^{(k)}\right) \text{ for } k = 1, \ldots, K_t, \qquad \mathcal{M}_t = \mathcal{M}_{t-1}^{(K_t)},$$

After processing all the chunks in $\mathcal{C}$, we obtain the final memory $\mathcal{M}_n$. Then we can calculate the rewards according to the final memory $\mathcal{M}_n$ and all the actions $\mathcal{A} = \{a_1, \cdots, a_n\}$ across the whole list of chunks.

### 3.1.2 REWARD FUNCTIONS

**Correctness Reward ($r_1$)** The correctness reward evaluates the comprehensiveness of the final memory $\mathcal{M}_n$ through question-answering performance. Given questions $\mathcal{Q} = \{q_1, \ldots, q_m\}$ and predicted answers $\mathcal{ANS} = \{ans_1, \ldots, ans_m\}$ obtained via the RAG pipeline, we compute $r_1$ using dataset-specific metrics (Table 6). For example, on SQuAD: $r_1 = l/m$ where $l$ is the number of correctly answered questions.

**Tool Call Format Reward ($r_2$)** To ensure reliable function execution, we reward tool calls with the correct format. For each function call $a_t^{(k)}$, let $s(a_t^{(k)}) \in \{0, 1\}$ be a binary indicator where $s(a_t^{(k)}) = 1$ if $a_t^{(k)}$ has the correct format and executes successfully and $0$ otherwise. The reward is: $r_{2,t} = \sum_{k=1}^{K_t} s(a_t^{(k)})/K_t$, measuring the percentage of successfully executed function calls.

**Compression Reward ($r_3$)** To encourage efficient memory usage, we define: $r_3 = 1 - l_m/l_c$, where $l_m$ is the total memory length and $l_c$ is the total length of the chunks. This promotes compact memory representations while preserving essential information.

**Memory Content Reward ($r_4$)** To ensure memory operations satisfy their semantic definitions, we use Qwen3-32b to validate memory updates (prompts in Appendix C.3). For each operation $a_t^{(k)}$, let $v(a_t^{(k)}) \in \{0, 1\}$ be a binary indicator where $v(a_t^{(k)}) = 1$ if $a_t^{(k)}$ is semantically valid and $0$ otherwise. The reward is: $r_{4,t} = \sum_{k=1}^{K_t} v(a_t^{(k)})/K_t$, measuring the fraction of valid operations.

Formal mathematical definitions of all reward components are provided in Appendix B.

Then we combine four rewards together to obtain the final reward $r_t$ for action $a_t$:

$$r_t = r_1 + r_{2,t} + \beta r_3 + \gamma r_{4,t} \tag{1}$$

where $\beta, \gamma$ are hyperparameters requiring tuning. We fix the weight of $r_{2,t}$ at 1 (rather than varying it) because the function call success rate is critical for memory updates. The four reward components operate at different granularities: $r_1$ (correctness) and $r_3$ (compression) are computed globally based on the final memory state $\mathcal{M}_n$ and thus share the same value across all actions in the sequence. In contrast, $r_{2,t}$ (tool call success) and $r_{4,t}$ (memory content quality) are evaluated at the action level, with each action $a_t = (a_t^{(1)}, \cdots, a_t^{(K_t)}), t \in \{1, \cdots, n\}$ receiving its own specific reward values based on the success rate of its function calls and the quality of its memory updates.

### 3.1.3 MEMORY COMPREHENSIVENESS EVALUATION VIA RAG

As outlined in Section 3.1.2, the comprehensiveness of the learned memory is evaluated by a decoupled retrieval-augmented generation (RAG) pipeline, where only the write policy is learnable and both retrieval and generation components remain fixed. After processing all context chunks, the agent outputs the terminal memory state $\mathcal{M}_n$. For each question $q_j$, evaluation proceeds in three stages: (1) **Retrieval**: For both semantic memory and episodic memory in $\mathcal{M}_n$, we use a fixed retriever $\phi$ that selects the top-$k$ memory entries from the corresponding memory pool using the BM25 retriever. (2) **Generation**: A frozen generator $g$ receives $q_j$ and the retrieved support set and produces an answer $ans'_j = g(q_j, \phi(\mathcal{M}_n, q_j))$. The system prompt is presented in Appendix C.3. (3) **Scoring**: We compare $ans'_j$ with the reference $ans_j$ to obtain correctness indicators, which induce the correctness reward $r_1$ described in Section 3.1.2.

### 3.2 POLICY OPTIMIZATION

We employ Group Relative Policy Optimization (GRPO) (Shao et al., 2024). In section 3.1.2, we eventually obtain the rewards for each action $a_t$ at step $t \in \{1, \cdots, n\}$. The advantage is:

$$A_t = A(\mathcal{M}_t, c_t, a_t) = \frac{r_t - \mu_{\text{group}}}{\sigma_{\text{group}} + \epsilon} = \frac{(r_1 + r_{2,t} + \beta r_3 + \gamma r_{4,t}) - \mu_{\text{group}}}{\sigma_{\text{group}} + \epsilon},$$

where $r_t$ is the obtained final reward for $a_t$ which consists of four different rewards. Then $\mu_{\text{group}}$ and $\sigma_{\text{group}}$ are the mean and standard deviation of rewards within the sampled action group, and $\epsilon$ is a small constant for numerical stability. The objective of Mem-$\alpha$ is to maximize the expected reward over all actions in the sequence:

$$\mathcal{J}(\theta) = \mathbb{E}_{\mathcal{C} \sim P(\mathcal{C}), \mathcal{A} \sim \pi_{\text{old}}(\cdot|\mathcal{C}, \mathcal{M}_0)} \sum_{t=1}^{n} \left[ \frac{1}{G} \sum_{i=1}^{G} \frac{1}{|a_t|} \sum_{j=1}^{|a_t|} \min\left( \frac{\pi_\theta(a_{t,j}|\mathcal{M}_t, c_t, a_{t,<j})}{\pi_{\text{old}}(a_{t,j}|\mathcal{M}_t, c_t, a_{t,<j})} A_t, \right. \right.$$

$$\left. \left. \text{clip}\left( \frac{\pi_\theta(a_{t,j}|\mathcal{M}_t, c_t, a_{t,<j})}{\pi_{\text{old}}(a_{t,j}|\mathcal{M}_t, c_t, a_{t,<j})}, 1 - \epsilon, 1 + \epsilon \right) A_t \right) \right], \tag{2}$$

where $\mathcal{C}$ is a list of context chunks, and $P(\mathcal{C})$ is the total set of possible lists. $\mathcal{M}_0$ is the initial empty memory, and $\mathcal{A}$ is the obtained actions from the chunks $\mathcal{C}$ and the initial memory $\mathcal{M}_0$. We discard the KL term in GRPO to encourage policy exploration.

### 3.3 MEMORY INSTANTIATION

We design a memory architecture comprising three complementary components, each serving distinct functional roles in long-term information management. (1) **Core Memory**: Following MemGPT (Packer et al., 2023), we maintain a persistent text summary (maximum 512 tokens) that remains continuously accessible in the agent's context. This component serves as a condensed representation of the most critical information, providing immediate access to essential context without retrieval overhead. (2) **Semantic Memory**: This component stores factual knowledge and declarative information about the world and user (Li & Li, 2024). We implement semantic memory as a structured collection of discrete factual statements, where each entry represents an atomic piece of knowledge that can be independently retrieved and updated. (3) **Episodic Memory**: This component captures temporally-grounded events and experiences (Li & Li, 2024; Liu et al., 2025; Anokhin et al., 2024; Pink et al., 2025; Fountas et al., 2024). We implement episodic memory as a chronologically-organized collection of timestamped events, enabling the agent to maintain temporal context and reconstruct interaction histories. Figure 3 illustrates the complete memory

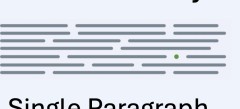

| Core Memory | Semantic Memory | Episodic Memory |
|---|---|---|
| ≡≡≡≡≡
≡≡≡≡≡
≡≡≡≡≡

Single Paragraph
(< 512 tokens) | • *Sky is blue.*
• *Birds can fly.*
• *Whales are mammals.*
...
Facts & Statements | • [9:15] Met Alice at café.
• [10:30] Attended team meeting.
...
Events with Timestamps |

Figure 3: **Memory Architecture**: Core Memory stores a single paragraph (max 512 tokens), while Semantic Memory and Episodic Memory maintain expandable lists of sentences for facts and timestamped events, respectively.

architecture and the interactions between these components. Each memory component is equipped with specialized operations tailored to its functional requirements. Semantic and episodic memories support fine-grained manipulation through three operations: `memory_insert` (adding new entries), `memory_update` (modifying existing entries), and `memory_delete` (removing entries). In contrast, core memory supports only `memory_update`, requiring complete rewriting to maintain coherence in its condensed representation. This design reflects the different update patterns: while semantic and episodic memories benefit from incremental modifications, core memory requires holistic revision to preserve its summarization quality. Importantly, our memory architecture is modular and decoupled from the reinforcement learning framework. Researchers can seamlessly substitute alternative memory designs—whether simpler or more complex—without modifying the training methodology, enabling flexible adaptation to diverse application requirements.

### 3.4 TRAINING DATASET PREPARATION

MemoryAgentBench (Hu et al., 2025) evaluates memory agents across four dimensions: (1) **Accurate Retrieval**: extracting correct information from historical data to address queries, encompassing both single-hop and multi-hop retrieval scenarios; (2) **Test-Time Learning**: acquiring new behaviors or capabilities during deployment; (3) **Long-Range Understanding**: integrating information distributed across multiple segments to answer queries requiring comprehensive sequence analysis; and (4) **Conflict Resolution**: revising, overwriting, or removing previously stored information when encountering contradictory evidence. Our work focuses on the first three dimensions, excluding Conflict Resolution due to the lack of realistic evaluation benchmarks—existing datasets for this dimension remain predominantly synthetic and do not adequately capture real-world complexity. We compile a training dataset comprising 4,139 instances, with detailed statistics presented in Table 6. Each instance consists of multiple context chunks, each of which triggers a distinct write action, resulting in long action sequences per instance. Given the computational overhead of reinforcement learning and the significant class imbalance in the full dataset, we employ a stratified sampling approach to create a balanced subset of 562 instances. The resulting distribution is detailed in Table 7, with comprehensive dataset preprocessing procedures described in Appendix A.1.

## 4 EXPERIMENTS

### 4.1 EXPERIMENTAL SETUP

**Evaluation Datasets and Metrics**   We follow MemoryAgentBench (Hu et al., 2025) and select representative datasets from three categories to comprehensively evaluate our approach: (1) Accurate Retrieval: We use Single-Doc, Multi-Doc and LME(S*) as the evaluation tasks. (2) Test-Time Learning: We evaluate on five multi-class classification datasets: TREC-C, TREC-F, NLU, CLINIC, BANKING77. (3) Long-Ran-Understanding, we use InfBench-Sum as the summarization task for evaluation. The detailed introduction of these datasets is in Appendix A.2.

**Baselines**   We compare with the following baselines: (1) Long-Context: We simply use Qwen3-32B as the long-context model. In our experiments, this model always has the maximum context window as 32k. (2) RAG-Top2: We use BM25 as the retrieval method, and use the question as the query, retrieve top two chunks from all the previous chunks, and then use Qwen3-32B as the model

| Method | Metric | AR | | | TREC-C | TTL | | LRU | Avg. |
|---|---|---|---|---|---|---|---|---|---|
| | | SQuAD | HotpotQA | PerLTQA | | NLU | Pubmed | BookSum | |
| Long-Context | Perf. | 0.742 | **0.852** | 0.605 | 0.623 | **0.708** | 0.533 | 0.052 | 0.588 |
| | Mem. | 10.6K | 9.7K | 13.1K | 3.9K | 6.1K | 16.7K | 15.4K | 10.8K |
| RAG-Top2 | Perf. | 0.762 | 0.849 | 0.623 | 0.612 | 0.508 | **0.570** | 0.042 | 0.567 |
| | Mem. | 10.6K | 9.7K | 16.7K | 3.9K | 6.1K | 16.7K | 15.6K | 11.3K |
| MemAgent | Perf. | 0.091 | 0.140 | 0.052 | 0.562 | 0.290 | 0.343 | 0.103 | 0.236 |
| | Mem. | 0.79K | 0.76K | 0.29K | 1.24K | 0.99K | 0.94K | 0.59K | 0.84K |
| MEM1 | Perf. | 0.039 | 0.083 | 0.068 | 0.269 | 0.056 | 0.175 | 0.085 | 0.111 |
| | Mem. | 0.16K | 0.22K | 0.14K | 0.23K | 0.22K | 0.08K | 0.16K | 0.17K |
| Mem-$\alpha$ | Perf. | **0.786** | 0.832 | **0.659** | **0.666** | 0.658 | 0.545 | 0.187 | **0.642** |
| | Mem. | 10.1k | 8.7k | 11.2k | 4.0k | 6.5k | 12.3k | 2.2k | 7.9k |

Table 1: Performance and the total number of tokens in the memory across validation datasets. **Perf.**: task-specific metrics (F1/Accuracy), **Mem.**: memory in thousands of tokens. AR: Accurate Retrieval, TTL: Time Time Learning, LRU: Long Range Understanding. Same as below.

to answer the questions. (3) MemAgent: We give the agent the specific task description, and then let the agent go over all the chunks, then ask the question according to the accumulated memory. (4) MEM1: Given all the chunks, the agent is required to maintain a paragraph of memory, retrieve some chunks, update the memory, and then answer the question according to the memory. The implementation details of the baselines are shown in Appendix C.2.

**Implementation Details**  Here we present the implementation details of Mem-$\alpha$ to ensure reproducibility. We use `verl` framework, choose Qwen3-4B as the backbone model[2], train on 32 H100 GPUs with learning_rate as 1e-6, batch_size as 32, grpo_rollout_n as 8 for three days. The complete training is 205 steps and we choose the best checkpoint according to the validation performance. In the main experiments, we choose the hyperparameters in Eq.(1) as $\beta = 0.05, \gamma = 1$. We show the performance variations with respect to different hyperparameter configurations in Section 4.4.

## 4.2 OVERALL PERFORMANCE COMPARISON

We present performance comparisons on validation datasets (matching the training distribution) in Table 1 and out-of-distribution test datasets (MemoryAgentBench) in Table 2. Our analysis yields four key findings: (1) Superior performance across tasks: Our method significantly outperforms existing baselines across all metrics. On MemoryAgentBench (Table 2), we observe particularly substantial improvements on Accurate Retrieval (AR) and Long-Range Understanding (LRU) tasks, demonstrating robust generalization to unseen distributions. (2) Efficient memory compression: Compared to Long-Context and RAG-Top2, our approach reduces memory footprint by approximately 50(3) Structured memory architecture matters: The limited performance of flat memory baselines (MEM1 and MemAgent), which employ single-paragraph representations, highlights the inadequacy of unstructured memory for complex information processing. This performance gap validates our hierarchical memory design and reinforcement learning-based optimization strategy. (4) Strong length generalization: Despite training exclusively on documents averaging <20K tokens, our method successfully generalizes to documents exceeding 400K tokens (up to 474K in MemoryAgentBench's Multi-Doc dataset), demonstrating the robustness of our training framework to extreme length extrapolation.

## 4.3 PERFORMANCE BOOST FROM REINFORCEMENT LEARNING

To demonstrate that the performance improvements in Section 4.2 stem from our reinforcement learning approach rather than merely the memory structure, we conduct ablation studies comparing three configurations: (1) our RL-tuned model with RL framework Mem-$\alpha$, (2) the base Qwen3-4B model with our memory framework, and (3) `gpt-4.1-mini` with our memory framework. Table 3 presents the validation dataset results. The base Qwen3-4B model achieves only 0.389 average performance—substantially below both RAG-Top2 (0.567) and Long-Context (0.588) from Table 1. While `gpt-4.1-mini` demonstrates stronger baseline performance (leveraging its superior instruction-following capabilities), our RL-tuned Mem-$\alpha$ achieves the highest performance, sur-

[2]We also tried Qwen3-8B but the performances are not as good, see details in Appendix C.1.

| Method | Metric | AR | | | TTL | | | | | LRU | Avg. |
|---|---|---|---|---|---|---|---|---|---|---|---|
| | | Single-Doc | Multi-Doc | LME(S) | TREC-C | NLU | TREC-F | Clinic | Banking77 | InfBench | |
| Long-Context | Perf. | 0.280 | 0.270 | 0.292 | 0.640 | **0.740** | 0.340 | **0.860** | **0.770** | 0.125 | 0.461 |
| | Mem. | 33K | 33K | 33K | 33K | 33K | 33K | 33K | 33K | 33K | 33K |
| RAG-Top2 | Perf. | 0.690 | 0.450 | **0.581** | 0.690 | 0.650 | 0.210 | 0.700 | 0.750 | 0.065 | 0.502 |
| | Mem. | 217K | 474K | 348K | 124K | 134K | 126K | 131K | 128K | 181K | 207K |
| MemAgent | Perf. | 0.070 | 0.160 | 0.050 | 0.370 | 0.260 | 0.210 | 0.250 | 0.370 | 0.043 | 0.198 |
| | Mem. | 1.02K | 1.02K | 0.56K | 1.02K | 1.02K | 0.77K | 1.02K | 1.02K | 0.73K | 0.92K |
| MEM1 | Perf. | 0.070 | 0.180 | 0.090 | 0.180 | 0.000 | 0.000 | 0.090 | 0.000 | 0.029 | 0.071 |
| | Mem. | 0.30K | 0.38K | 0.22K | 0.16K | 0.11K | 0.13K | 0.28K | 0.11K | 0.19K | 0.21K |
| Mem-$\alpha$ | Perf. | **0.740** | **0.680** | 0.520 | **0.710** | 0.710 | **0.410** | 0.730 | 0.700 | **0.129** | **0.592** |
| | Mem. | 160K | 323K | 127K | 120K | 142K | 123K | 18K | 133K | 19K | 129K |

Table 2: Performance and the total number of tokens in the memory on MemoryAgentBench. **Perf.**: task-specific metrics (F1/Accuracy), **Mem.**: memory in thousands of tokens.

| Method | Metric | AR | | | TTL | | | LRU | Avg. |
|---|---|---|---|---|---|---|---|---|---|
| | | SQuAD | HotpotQA | PerLTQA | TREC-C | NLU | Pubmed | BookSum | |
| Qwen3-4B | Perf. | 0.338 | 0.637 | 0.557 | 0.416 | 0.381 | 0.281 | 0.130 | 0.389 |
| | Mem. | 3.3K | 4.8K | 9.0K | 2.3K | 2.9K | 4.4K | 0.9K | 3.9K |
| gpt-4.1-mini | Perf. | 0.426 | 0.749 | 0.492 | 0.637 | 0.519 | 0.544 | **0.246** | 0.517 |
| | Mem. | 3.8K | 4.9K | 3.7K | 3.4K | 5.9K | 10.6K | 1.5K | 4.8K |
| Qwen3-4B w/ | Perf. | **0.786** | **0.832** | **0.659** | **0.666** | **0.658** | **0.545** | 0.187 | **0.642** |
| Mem-$\alpha$ | Mem. | 10.1K | 8.7K | 11.2K | 4.0K | 6.5K | 12.3K | 2.2K | 7.9K |

Table 3: Performance and memory consumption comparison across evaluation datasets. **Perf.**: task-specific metrics (F1/Accuracy), **Mem.**: memory in thousands of tokens. All methods use BM25 retrieval with `qwen3-32b`. Bold indicates best results.

passing even `gpt-4.1-mini`. These results provide compelling evidence that our performance gains originate from the reinforcement learning optimization rather than the memory architecture alone. The dramatic improvement from base Qwen3-4B (0.389) to Mem-$\alpha$ (0.642) demonstrates that our RL framework successfully trains the model to effectively utilize the memory structure, transforming a relatively weak base model into a state-of-the-art memory-augmented agent.

## 4.4 ABLATION STUDIES

Our reward function, defined in Eq. (1), comprises four components: $r_1$ (accuracy), $r_2$ (tool call format), $r_3$ (compression), and $r_4$ (memory content quality). We fix the weights of the primary components $r_1$ and $r_2$ to 1.0, as they directly measure task performance, and tune only the compression weight $\beta$ and memory content weight $\gamma$. Our experiments employ $\beta = 0.05$ and $\gamma = 0.1$ as default values. Table 4 presents ablation studies (The results on the test dataset MemoryAgent-Bench is shown in Appendix C.4.) examining the impact of these hyperparameters, yielding two key findings. First, the memory content reward ($\gamma$) proves critical for effective learning: setting $\gamma = 0$ leads to catastrophic performance degradation, as the model fails to acquire meaningful memory construction strategies, resulting in disorganized memory representations that cannot support downstream tasks. Second, the compression reward ($\beta$) exhibits task-dependent effects. While maintaining $\gamma = 0.1$, increasing $\beta$ produces shorter memories at the cost of reduced performance. Notably, comparing configurations ($\beta = 0.05, \gamma = 0.1$) and ($\beta = 0, \gamma = 0.1$), we observe substantial memory reduction on BookSum (2.2K vs. 4.5K tokens) while maintaining comparable memory lengths on other datasets. This demonstrates that our chosen configuration ($\beta = 0.05, \gamma = 0.1$) achieves an optimal balance between memory efficiency and task performance.

## 4.5 CASE STUDIES

In this section, we report some memory construction traces obtained from Mem-$\alpha$ and compare them with baseline approaches to demonstrate the effectiveness of our memory management strategy. Table 5 illustrates critical differences in how different models handle memory construction. Qwen3-4B exhibits severe limitations: it fails to update the core memory entirely (leaving it empty), and only maintains a single semantic memory entry, resulting in significant information loss as multiple distinct concepts are compressed into one generic statement. GPT-4.1-mini demonstrates better semantic organization with three distinct entries, but suffers from inefficient episodic memory man-

| $\beta$ | $\gamma$ | Metric | SQuAD | AR HotpotQA | PerLTQA | TREC-C | TTL NLU | Pubmed | LRU BookSum | Avg. |
|---|---|---|---|---|---|---|---|---|---|---|
| 0.05 | 0.0 | Perf. | 0.701 | 0.802 | 0.652 | 0.423 | 0.542 | 0.501 | 0.183 | 0.543 |
| | | Mem. | 9.2K | 8.2K | 10.8K | 3.0K | 3.5K | 11.0K | 4.9K | 7.5K |
| 0.0 | 0.1 | Perf. | 0.817 | **0.853** | **0.678** | 0.605 | 0.629 | **0.572** | 0.183 | 0.630 |
| | | Mem. | 9.7K | 8.1K | 11.7K | 3.7K | 5.4K | 12.5K | 4.5K | 7.9K |
| 0.05 | 0.1 | Perf. | 0.786 | 0.832 | 0.659 | **0.666** | **0.658** | 0.545 | 0.187 | **0.642** |
| | | Mem. | 10.1K | 8.7K | 11.2K | 4.0K | 6.5K | 12.3K | 2.2K | 7.9K |
| 0.2 | 0.1 | Perf. | **0.822** | 0.838 | 0.615 | 0.558 | 0.176 | 0.401 | 0.193 | 0.525 |
| | | Mem. | 9.8K | 7.8K | 10.4K | 0.4K | 0.8K | 0.4K | 3.0K | 4.7K |
| 0.4 | 0.1 | Perf. | 0.691 | 0.810 | 0.533 | 0.475 | 0.405 | 0.455 | **0.201** | 0.509 |
| | | Mem. | 8.8K | 8.1K | 5.2K | 0.7K | 1.4K | 1.3K | 1.5K | 3.6K |

Table 4: Performance and memory consumption comparison across evaluation datasets. **Perf.**: task-specific metrics (F1/Accuracy), **Mem.**: memory in thousands of tokens. All methods use BM25 retrieval with `qwen3-32b`. Bold indicates best results.

| Memory Type | Qwen3-4B | GPT-4.1-mini | Qwen3-4B w/ Mem-$\alpha$ |
|---|---|---|---|
| **Core** | ∅ ✗ Should not be empty | User is ... focusing on minimizing noise pollution ... currently looking for condos, particularly in downtown areas ... ✓ | User is looking to get some advice on condo living ... looking at options for condo in the downtown area ... ✓ |
| **Semantic** | User is seeking advice on ... noise pollution ... amenities. ✗ Should record more | *3 distinct entries:* 
 - Noise pollution tips 
 - Neighborhood evaluation 
 - Research importance 
 (✓ Complete) | *2 distinct entries:* 
 - Noise proof tips ... 
 - Research methods ... 
 (✓ Complete) |
| **Episodic** | At 2023/03/08 01:55, User asked ... Assistant provided ... (✓ Concise and Complete) | At 2023/03/08 01:55, Asked for noise tips 
 At 2023/03/08 01:55, Requested neighborhood eval 
 At 2023/03/08 01:55, Inquired about research 
 ✗ Multiple events with same timestamps, can be consolidated; Only records user behavior, missing all assistant behaviors. | At 2023/03/08 (Wed) 01:55 user looked to get some advice on condo living... assistant responded with ... (✓ Concise and Complete) |

Table 5: Comparison of Memory Management Strategies Across Models

agement by creating multiple entries with identical timestamps that should be merged to conserve memory space. Meanwhile, GPT-4.1-mini is only storing the user behavior, completely ignoring the responses from the assistant. In contrast, Mem-$\alpha$ demonstrates better memory construction by maintaining informative core memory, organizing semantic information into detailed, distinct entries, efficiently consolidating episodic events with the same timestamp into a single comprehensive entry, paying attention to both the user behavior and the assistant response. This superior memory organization enables Mem-$\alpha$ to retain more information while using memory space more efficiently.

## 5 CONCLUSION, LIMITATION AND FUTURE WORK

In this work, we presented Mem-$\alpha$, a reinforcement learning framework that enables LLM agents to learn effective memory management strategies through interaction and feedback. By moving beyond pre-defined heuristics, our approach allows agents to discover optimal memory operations for diverse scenarios through a carefully designed training dataset and reward mechanism based on question-answering correctness. Our experiments demonstrate that Mem-$\alpha$ achieves significant improvements over existing memory-augmented baselines, with agents developing robust memory management strategies that generalize well to much longer interaction patterns. While our framework shows strong performance, several promising directions remain for future exploration. Our current memory architecture could benefit from integration with more sophisticated systems like MIRIX, which may provide additional structural advantages for complex reasoning tasks. Furthermore, extending Mem-$\alpha$ from simulated environments to real-world applications would require connecting the reinforcement learning framework with actual databases and production systems, introducing challenges around latency, scalability, and safety that warrant careful investigation. These directions represent exciting opportunities to bridge the gap between learned memory management and practical deployment of memory-augmented LLM agents in real-world applications.

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

# A DATASETS DETAILS

## A.1 TRAINING DATASET

We organize our training data into three categories based on the memory capabilities they target, as illustrated in Section 3.4. The detailed dataset statistics are provided in Table 6.

| Dataset | Cat. | Metric | Training Set | | | | Validation Set | | | |
|---|---|---|---|---|---|---|---|---|---|---|
| | | | Ins. | Tok/Ch | Ch/Ins | Q/Ins | Ins. | Tok/Ch | Ch/Ins | Q/Ins |
| SQuAD | AR | SubEM | 264 | 1,078 | 10.0 | 95.5 | 30 | 1,057 | 10.0 | 96.8 |
| HotpotQA | AR | SubEM | 1,966 | 1,051 | 9.3 | 17.0 | 219 | 1,052 | 9.2 | 17.0 |
| PerLTQA | AR | SubEM | 27 | 517 | 23.3 | 100.0 | 4 | 568 | 23.0 | 100.0 |
| LME-Train | AR | LLM-J | 45 | 1,522 | 15.6 | 4.0 | 5 | 1,576 | 13.4 | 4.0 |
| NLU | TTL | EM | 180 | 610 | 10.0 | 100.0 | 20 | 606 | 10.0 | 100.0 |
| TREC-C | TTL | EM | 180 | 390 | 10.0 | 100.0 | 20 | 390 | 10.0 | 100.0 |
| PubMed | TTL | EM | 90 | 1,676 | 10.0 | 100.0 | 10 | 1,673 | 10.0 | 100.0 |
| BookSum | LRU | KW Hit | 1,387 | 1,916 | 8.0 | 1.0 | 155 | 1,914 | 8.1 | 1.0 |
| **Total** | | | **4,139** | – | – | – | **463** | – | – | – |

Table 6: Dataset statistics across 8 data sources. Each dataset is evaluated with specific metrics suitable for its task type. Column abbreviations: Cat. = Category (AR: Accurate Retrieval, TTL: Test-Time-Learning, LRU: Long Range Understanding); Ins. = Number of Instances; Tok/Ch = Average Tokens per Chunk; Ch/Ins = Average Chunks per Instance; Q/Ins = Average Questions per Instance.

**Accurate Retrieval (AR)**  This category focuses on training the model's ability to store and precisely retrieve information from memory. We employ the following datasets:

(1) **SQuAD** (Rajpurkar et al., 2016): We adapt this single-document question answering dataset by combining multiple documents into single instances. The agent must memorize these documents and subsequently answer questions based on the constructed memory, testing its ability to accurately retrieve specific information.

(2) **HotPotQA** (Yang et al., 2018): This multi-document question answering dataset presents the agent with sequential chunks, each potentially containing multiple documents. The agent must memorize the documents, identify relationships between them, and answer questions requiring information synthesis across independent chunks.

(3) **PerLTQA** (Du et al., 2024): This dataset challenges the agent to reason over memory chunks containing both episodic and semantic information about users. The agent must identify relevant memories, integrate information across different memory types, maintain user profile consistency, and perform multi-hop reasoning to answer questions.

(4) **LongMemEval-Train** (Wu et al., 2024): We construct a training subset from LongMemEval by collecting 200 questions from `longmemeval_oracle.json`[3], ensuring no overlap with the evaluation data in MemoryAgentBench. We concatenate haystack dialogues into contexts ranging from 10K to 30K tokens, with each context paired with 4-5 questions, resulting in 50 training samples.

**Test-Time Learning (TTL)**  This category trains the model's ability to learn new classification patterns from examples and apply them to new instances. We employ the following datasets:

(1) **PubMed-RCT** (Dernoncourt & Lee, 2017): We adapt this large-scale dataset of randomized controlled trial abstracts from medical literature for test-time learning. Each sentence is originally annotated with semantic roles (Background, Objective, Method, Result, or Conclusion). We transform this into a classification learning task by segmenting the data into conversational chunks containing multiple sentence-label pairs as training examples. To evaluate the agent's ability to learn abstract patterns, we replace semantic labels with numeric labels (0-4). Each instance ensures coverage of all five categories across chunks, with questions prompting classification of new examples.

---

[3](https://huggingface.co/datasets/xiaowu0162/longmemeval/tree/main)

| Dataset | Cat. | Metric | Training Set | | | | Validation Set | | | |
|---|---|---|---|---|---|---|---|---|---|---|
| | | | Ins. | Ch/Ins | Tok/Ch | Q/Ins | Ins. | Ch/Ins | Tok/Ch | Q/Ins |
| SQuAD | AR | SubEM | 100 | 9.9 | 1,084.1 | 94.8 | 30 | 10.0 | 1,057.0 | 96.8 |
| HotpotQA | AR | SubEM | 100 | 9.7 | 1,005.4 | 16.7 | 219 | 9.2 | 1,051.6 | 17.0 |
| PerLTQA | AR | SubEM | 27 | 23.3 | 517.1 | 100.0 | 4 | 23.0 | 567.8 | 100.0 |
| LME-Train | AR | LLM-J | 50 | 15.4 | 1527.7 | 4.0 | - | - | - | - |
| NLU | TTL | EM | 49 | 10.0 | 610.9 | 100.0 | 20 | 10.0 | 606.2 | 100.0 |
| TREC-Coarse | TTL | EM | 51 | 10.0 | 390.1 | 100.0 | 20 | 10.0 | 390.2 | 100.0 |
| PubMed-RCT | TTL | EM | 90 | 10.0 | 1,676.1 | 100.0 | 10 | 10.0 | 1,673.3 | 100.0 |
| BookSum | LRU | KW Hit | 100 | 7.8 | 1,909.7 | 1.0 | 155 | 8.1 | 1,914.3 | 1.0 |
| **Total** | | | **562** | – | – | – | **463** | – | – | – |

Table 7: Dataset statistics across 8 data sources. Each dataset is evaluated with specific metrics suitable for its task type. Column abbreviations: Cat. = Category (AR: Accurate Retrieval, TTL: Test-Time-Learning, LRU: Long Range Understanding); Ins. = Number of Instances; Tok/Ch = Average Number of Tokens per Chunk; Ch/Ins = Average Number of Chunks per Instance; Q/Ins = Average Questions per Instance.

(2) **NLU and TREC-C**: These datasets are adapted from MemoryAgentBench (Hu et al., 2025), containing documents with labeled sentences across 68 classes (NLU) and 6 classes (TREC-C). Given the original instances contain approximately 100K tokens, we partition them into manageable chunks. We create 200 instances per dataset, each containing 10 chunks with roughly $500 \sim 2,000$ tokens distributed across chunks. Each instance preserves all original labels while redistributing training examples to ensure complete label coverage within each instance.

**Long Range Understanding (LRU)**    This category focuses on training the model's ability to comprehend and summarize information across extended contexts. We employ the following dataset:

**BookSum** (Kryściński et al., 2021): We utilize the cleaned version of this dataset[4], where each item consists of a book chapter paired with its summary. We segment each chapter into 10-20 conversational chunks to simulate incremental information processing. For evaluation, we extract keywords from ground-truth summaries using the prompt shown in Figure 4. The evaluation metric is the ratio of correctly identified keywords in generated summaries compared to the ground-truth keyword set.

Due to computational constraints and dataset imbalance, we limit each dataset to a maximum of 100 instances. Despite training for three days with 32 H100 GPUs, we could only process a small portion of the complete datasets. The final dataset composition and statistics are presented in Table 7. We process every chunk into the format of conversations, with the examples or formats of each dataset shown in Figure 5.

## A.2 EVALUATION DATASET

To comprehensively evaluate our model's memory capabilities across different scenarios, we adopt the evaluation framework from MemoryAgentBench (Hu et al., 2025) and select representative datasets from three core categories. This evaluation suite encompasses 9 datasets with 112 test instances, designed to assess accurate retrieval, test-time learning, and long-range understanding capabilities. The detailed statistics for each dataset are presented in Table 8.

**Accurate Retrieval (AR)**    This category evaluates the model's ability to precisely locate and retrieve specific information from memory. We employ the following datasets:

(1) **RULER-QA1 (Single-Hop)** and **RULER-QA2 (Multi-Hop)**: These datasets test single-hop and multi-hop question answering capabilities respectively. RULER-QA1 (Hsieh et al., 2024) requires direct information retrieval, while RULER-QA2 demands reasoning across multiple memory chunks to synthesize answers.

---

[4]https://huggingface.co/datasets/ubaada/booksum-complete-cleaned

> **The prompt used to extract keywords in the summaries of BookSum and InfBench-Sum**
>
> Analyze the following book summary and extract the most important keywords. Focus on:
>
> 1. Character names (main and supporting characters)
> 2. Key events and plot points
> 3. Important locations/settings
> 4. Central themes and concepts
> 5. Significant objects or symbols
> 6. Time periods or dates mentioned
> 7. Key relationships between characters
> 8. Important actions or decisions
>
> Example:
> Summary: "Elizabeth Bennet meets Mr. Darcy at a ball in Hertfordshire. Initially, she finds him proud and disagreeable. After learning about his past with Wickham and his role in separating Jane and Bingley, her dislike intensifies. However, when Darcy proposes and she rejects him, he writes a letter explaining his actions. Elizabeth realizes her prejudices and eventually falls in love with him after visiting Pemberley."
>
> Keywords: Elizabeth Bennet, Mr. Darcy, ball, Hertfordshire, proud, Wickham, Jane, Bingley, proposal, rejection, letter, prejudices, Pemberley, love, Pride and Prejudice themes, marriage, social class, first impressions, misunderstanding, character growth
>
> Now analyze this summary:
> ⟨Summary⟩
>
> Extract keywords/phrases that capture the essential information in this summary, make sure they are complete and cover all aspects of the story.
> Return ONLY a comma-separated list of keywords, nothing else.
> Focus on concrete, specific terms rather than generic words.
> Include both single words and short phrases (2-3 words max).
> Prioritize proper nouns, specific events, and unique concepts.

Figure 4: The prompt used to extract keywords in the summaries of BookSum and InfBench-Sum.

| Dataset | Category | Evaluation Metric | Test Set | | | |
|---|---|---|---|---|---|---|
| | | | # of Ins. | Avg. Chunks per Instance | Avg. Tokens per Chunk | Avg. Q's per Instance |
| Banking77 | ICL | Source-based | 1 | 111.0 | 1,150.3 | 100.0 |
| Clinic150 | ICL | Source-based | 1 | 38.0 | 3,440.5 | 100.0 |
| NLU | ICL | EM | 1 | 115.0 | 1,166.7 | 100.0 |
| TREC-Coarse | ICL | EM | 1 | 111.0 | 1,114.6 | 100.0 |
| TREC-Fine | ICL | EM | 1 | 108.0 | 1,163.3 | 100.0 |
| InfBench-Sum | LRU | Source-based | 100 | 88.9 | 2,034.1 | 1.0 |
| LongMemEval | AR | LLM judge | 5 | 218.6 | 1,591.4 | 60.0 |
| RULER-QA1 | AR | Source-based | 1 | 103.0 | 2,103.9 | 100.0 |
| RULER-QA2 | AR | Source-based | 1 | 219.0 | 2,163.5 | 100.0 |
| **Total** | | | **112** | – | – | – |

Table 8: Test dataset statistics across 9 data sources. Each dataset is evaluated with specific metrics suitable for its task type.

(2) **LME(S\*)**: Originally from LongMemEval (Wu et al., 2024), this dataset was processed by Hu et al. (2025) to create a more evaluation-efficient format where multiple questions are posed against fewer contexts, testing the model's ability to maintain and query complex memory representations over extended interactions.

**Test-Time Learning (TTL)**   This category assesses the model's ability to learn new classification patterns from examples and apply them to novel instances. The context used in this dataset includes thousands of labeled examples. Each example is labeled with a number to indicate the category. We employ the following datasets:

---

**Prompts Used for Memory Construction on Various Tasks**

***Document Question Answering (SQuAD or HotpotQA):***
Dialogue between User and Assistant on 2024-01-01 00:00:
⟨User⟩: I have some interesting updates for you:
The peninsular borough's maritime heritage ... one brother's gambling debts.
⟨Assistant⟩: Understood. I'll keep these facts for future reference.

***PerLTQA:***
The following is the event happened about the user Xiong Fei on 2017:
Summary: Sister is threatened
Content: In 2017, a constitutional dispute involving freedom of speech ... behind freedom of speech.

The following are the dialogues.

Dialogue happened at 2022-05-12 08:30:00
⟨Assistant⟩: Hello, how can I help you?
⟨Xiong Fei⟩: ...
⟨Assistant⟩: ...
...

***LME-Train:***
Dialogue at timestamp 2023/05/25 (Thu) 17:08
⟨User⟩: I'm looking to buy a house and ...
⟨Assistant⟩: Mortgage insurance (MI) can indeed ...

***Test-Time-Learning (Pubmed-RCT, NLU, Trec-C):***
Dialogue between User and Assistant on 2024-01-01 00:00
⟨User⟩: The following are classification examples with their corresponding labels:
⟨Sample: xxx; Label: xxx⟩
⟨Assistant⟩: Great! I've added this to my knowledge base.

***BookSum:***
Event happened on 2024-01-01 The user is reading a book
⟨User⟩: ⟨chunk⟩.
⟨System⟩: Please remember what the user reads on 2024-01-01, save the details within the book, and retain a summary of the book the user has read so far.

Figure 5: The examples in the training dataset. For SQuAD, HotpotQA, PerLTQA, LME-Train, we show the examples directly; for Test-Time-Learning datasets (Pubmed-RCT, NLU, and Trec-C) and BookSum, we demonstrate the format for clarity.

(1) **TREC-Coerse**: A question classification dataset with 6 broad categories, testing the model's ability to learn coarse-grained classification patterns from limited examples. The original dataset (Li & Roth, 2002) contains 5,452 training questions and 500 test questions and it is a standard benchmark for QA question-type classification.

(2) **TREC-Fine**: A fine-grained version with 50 specific question types, evaluating the model's capacity to distinguish between subtle classification boundaries. The original dataset (Li & Roth, 2002) keeps the same size (5,452 train / 500 test) but refines labels into 50 subtypes under the 6 top-level categories, increasing granularity for few-shot intent learning.

(3) **NLU**: A natural language understanding dataset with 68 intent categories, challenging the model to learn complex semantic patterns from conversational examples. The original released corpus has 25,715 utterances across 18 scenarios and 68 intents (Liu et al., 2019).

(4) **CLINIC150**: A medical intent classification dataset with 150 categories, testing domain-specific learning capabilities in healthcare scenarios. The official full split provides 150 in-scope intents across 10 domains with 100/20/30 train/validation/test examples per intent (Larson et al., 2019).

(5) **Banking77**: A financial services dataset with 77 intent categories, evaluating the model's ability to learn domain-specific classification patterns in banking contexts. Casanueva et al. (2020) comprises 13,083 customer-service queries (77 intents) with a 10,003/3,080 train/test split and targets fine-grained single-domain intent detection.

**Long Range Understanding (LRU)**   This category evaluates the model's ability to comprehend and synthesize information across extended contexts. We employ the following dataset:

**InfBench-Sum**: A summarization dataset from InfBench (Zhang et al., 2024), requiring the model to process long-form content across multiple chunks and generate coherent summaries. This tests the model's capacity to maintain contextual understanding over extended sequences and synthesize information from distributed memory representations. This dataset includes 100 novels, with an average context length of 172k tokens. During evaluation, the model is required to read a long novel and generate a corresponding high-level summary.

## B   FORMAL DEFINITIONS OF REWARD COMPONENTS

This section provides the formal mathematical definitions of the four reward components used in our reinforcement learning framework.

**Correctness Reward** ($r_1$)   Given a final memory state $\mathcal{M}_n$ after processing all chunks $\mathcal{C} = \{c_1, \ldots, c_n\}$, and a set of questions $\mathcal{Q} = \{q_1, \ldots, q_m\}$ with ground truth answers $\mathcal{R} = \{r_1, \ldots, r_m\}$, the correctness reward is defined as:

$$r_1 = \frac{1}{m} \sum_{j=1}^{m} \mathbb{I}[\text{metric}(\hat{r}_j, r_j)]$$

where $\hat{r}_j = g(q_j, \phi(\mathcal{M}_n, q_j))$ is the predicted answer generated by the RAG pipeline, $\text{metric}(\cdot, \cdot)$ is the dataset-specific evaluation metric (e.g., exact match, F1 score), and $\mathbb{I}[\cdot]$ is the indicator function.

**Tool Call Format Reward** ($r_2$)   For each time step $t \in \{1, \ldots, n\}$ with action $a_t = (a_t^{(1)}, \ldots, a_t^{(K_t)})$, define the tool call format correctness indicator:

$$s(a_t^{(k)}) = \begin{cases} 1 & \text{if function call } a_t^{(k)} \text{ executes without error} \\ 0 & \text{otherwise} \end{cases}$$

The tool call format reward at time step $t$ is:

$$r_{2,t} = \frac{1}{K_t} \sum_{k=1}^{K_t} s(a_t^{(k)})$$

**Compression Reward** ($r_3$)   Given the total length of input chunks $l_c = \sum_{i=1}^{n} |c_i|$ and the total memory length $l_m = |\mathcal{M}_n|$ (sum of all memory entries), the compression reward is:

$$r_3 = 1 - \frac{l_m}{l_c}$$

This reward encourages the agent to maintain compact memory representations while preserving essential information. The reward approaches 1 when memory is highly compressed and approaches 0 when memory size equals input size.

**Memory Content Reward** ($r_4$)   For each time step $t \in \{1, \ldots, n\}$ with action $a_t = (a_t^{(1)}, \ldots, a_t^{(K_t)})$, define the validity indicator using a language model judge:

$$v(a_t^{(k)}) = \begin{cases} 1 & \text{if operation } a_t^{(k)} \text{ is semantically valid per LM judge} \\ 0 & \text{otherwise} \end{cases}$$

The memory content reward at time step $t$ is:

> **Universal Prompt of Mem-$\alpha$**
>
> Remember the following content chunk by completing these steps:
>
> 1. **Core Memory Update**: Maintain an understanding of the user, or a summary of what the user is reading, or a set of classification rules summarized from the classification examples (label 1: meaning; label 2: meaning, etc.). Keep updates brief (a few sentences maximum).
>
> 2. **Memory Storage**:
> - **Episodic Memory**: Record user actions, user's friends' actions and assistant actions with timestamps (format: "At timestamp t, user did X")
> - **Semantic Memory**: Record key facts and information (format: "John is User's 18-year-old friend", "Harry Potter author: J.K. Rowling", "Sample: xxx; Label: xxx")
>
> <new_chunk>
> {context}
> </new_chunk>
>
> *Important**: Response limit is {max_new_tokens} tokens. Be concise and brief in all memory updates.

Figure 6: The universal prompt used in the training of Mem-$\alpha$.

$$r_{4,t} = \frac{1}{K_t} \sum_{k=1}^{K_t} v(a_t^{(k)})$$

The overall reward combines these components as: $r = r_1 + r_2 + \beta r_3 + \gamma r_4$, where $r_1$ and $r_3$ are global rewards shared across all time steps, while $r_2$ and $r_4$ are computed per time step.

## C EXPERIMENTAL DETAILS

### C.1 JUSTIFICATION OF BACKBONE MODEL SELECTION

We also evaluated Qwen3-8B but encountered critical instruction-following issues that made it unsuitable for our experiments. Despite explicit function signature specifications requiring the argument memory_type to accept only the values 'semantic', 'core', or 'episodic', Qwen3-8B consistently generated malformed function calls such as new_memory_insert(memory_type='semantic_memory'), appending an unnecessary '_memory' suffix to the argument values. This systematic failure to adhere to the specified API format occurred reliably across multiple trials. To investigate whether this was a formatting preference rather than a fundamental limitation, we modified our function signatures to accommodate the model's apparent preference, changing the valid arguments to 'semantic_memory', 'core_memory', and 'episodic_memory'. While this adaptation did not impact Qwen3-4B's performance (which handled both formats correctly), Qwen3-8B continued to exhibit lower reward values compared to Qwen3-4B even with this accommodation. This counterintuitive result—where the larger model demonstrated both poorer instruction-following capabilities and lower overall performance than the 4B variant—led us to exclude Qwen3-8B from our final experiments.

### C.2 BASELINE INTRODUCTION AND IMPLEMENTATION DETAILS

We compare with the following baselines:

(1) **Long-Context**: We simply use Qwen3-32B as the long-context model. In our experiments, this model always has the maximum context window as 32k. For the dataset with a total chunk length exceeding 32k, we truncate the combined chunk to keep the last 32k tokens.

(2) **RAG-Top2**: We use BM25 as the retrieval method, and use the question as the query, retrieve top two chunks from all the previous chunks, and then use Qwen3-32B as the model to answer the questions.

---

**The prompt used to measure the content of the Core Memory**

You are an expert memory analyst. Analyze the quality of core memory content.

The core memory is invalid if any of the following meets:
(1) The literal content "core memory" appears in the memory such as "This is core memory ...", "The core memory has been updated ...".
(2) The core memory is apparently a placeholder such as "Here we save the summary" while not stating what the "summary" is, "Here are some rules" and not stating what the "rules" are.

Otherwise, the core memory is valid.

Respond ONLY with a JSON code block in this exact format:
```json
{
"VALID": true/false,
"ISSUES": [list any problems found],
"EXPLANATION": "brief explanation of the assessment"
} ```

---

Figure 7: The prompt used to measure the content of the Core Memory during training.

(3) **MemAgent**: We adopt the code from `https://github.com/BytedTsinghua-SIA/MemAgent` and use the 14B version `BytedTsinghua-SIA/RL-MemoryAgent-14B` to construct the memory.

(4) **MEM1**: We use the code from `https://github.com/MIT-MI/MEM1` and use the model `https://huggingface.co/Mem-Lab/Qwen2.5-7B-RL-RAG-Q2-EM-Release` to construct the memory.

For both baselines MemAgent and MEM1, we let the model go over all the chunks $\mathcal{C}$ with the instruction including the task description, then with the obtained memory, we let the model answer questions. For MemAgent, we use the original model to answer the questions, for MEM1, after obtaining the memory, we use Qwen3-32B as the model to answer the questions based on the question and the obtained memory.

### C.3 PROMPTS USED IN TRAINING

**Instruction to Memorize the Chunk**  In our training, we use a universal prompt for the whole dataset, and we show the prompt in Figure 6. During update, when processing each chunk, we use this prompt to ask the agent to memorize the information in the chunk.

**Prompt to Measure Memory Content**  When computing the memory content reward $r_4$, we use the model Qwen3-32B as the judge. For Core Memory, Episodic Memory and Semantic Memory, we use the prompt in Figure 7, 8, 9, respectively.

**Prompt to Answer the Questions**  When using the final model Qwen3-32B to answer the questions, we use the prompt as shown in Figure 10.

### C.4 ADDITIONAL ABLATION STUDY

In Section 4.4, we show the performance comparison of different $\beta, \gamma$ on the validation dataset. We also compare these settings on the test dataset (MemoryAgentBench), shown in Table 9. The observations are consistent with Section 4.4.

---

**The prompt used to measure the content of the Episodic Memory**

You are an expert memory analyst. Analyze the quality of episodic memory content.

Episodic memory should contain:
- Experiences or events
- Clear temporal information (when it happened)
- Contextual details (what happened)

Respond ONLY with a JSON code block in this exact format:
```json
{
"VALID": true/false,
"ISSUES": [list any problems found],
"EXPLANATION": "brief explanation of the assessment"
}
```

Figure 8: The prompt used to measure the content of the Episodic Memory during training.

---

**The prompt used to measure the content of the Semantic Memory**

You are an expert memory analyst. Analyze the quality of semantic memory content.

Semantic memory should contain:
- Information or Knowledge about somebody or something
- Definitions, theories, principles, or explanations
- How-to knowledge or procedural information
- Research findings or established facts

Two other memories are Core memory (User Personalities) and Episodic memory (User Experiences). The information not suitable for these two memories should be considered as semantic memory.

Respond ONLY with a JSON code block in this exact format:
```json

"VALID": true/false,
"ISSUES": [list any problems found],
"EXPLANATION": "brief explanation of the assessment"
```

Figure 9: The prompt used to measure the content of the Semantic Memory during training.

| $\beta$ | $\gamma$ | Metric | AR | | | TTL | | | | | LRU | Avg. |
| | | | Single-Doc | Multi-Doc | LME(S) | TREC-C | NLU | TREC-F | CLINIC | BANKING77 | InfBench-Sum | |
| --- | --- | --- | --- | --- | --- | --- | --- | --- | --- | --- | --- | --- |
| 0.05 | 0.0 | Perf. | 0.420 | 0.340 | **0.527** | 0.480 | 0.640 | 0.200 | 0.720 | 0.550 | 0.108 | 0.445 |
| | | Mem. | 86K | 123K | 159K | 75K | 100K | 65K | 20K | 97K | 54K | 87K |
| 0.0 | 0.1 | Perf. | **0.770** | 0.610 | 0.387 | 0.690 | **0.730** | 0.370 | **0.780** | **0.770** | 0.109 | 0.580 |
| | | Mem. | 160K | 362K | 47K | 124K | 113K | 127K | 47K | 119K | 41K | 127K |
| 0.05 | 0.1 | Perf. | 0.740 | 0.680 | 0.520 | 0.710 | 0.710 | **0.410** | 0.730 | 0.700 | **0.129** | **0.592** |
| | | Mem. | 160K | 323K | 127K | 120K | 142K | 123K | 18K | 133K | 19K | 129K |
| 0.2 | 0.1 | Perf. | 0.710 | **0.730** | 0.367 | **0.810** | 0.270 | 0.280 | 0.140 | 0.020 | 0.113 | 0.351 |
| | | Mem. | 160K | 344K | 139K | 3K | 3K | 3K | 1K | 5K | 118K | 87K |
| 0.4 | 0.1 | Perf. | 0.590 | 0.610 | 0.453 | 0.500 | 0.360 | 0.190 | 0.170 | 0.380 | 0.119 | 0.375 |
| | | Mem. | 138K | 312K | 27K | 1K | 1K | 1K | 2K | 1K | 16K | 55K |

Table 9: Performance and memory consumption on MemoryAgentBench. **Perf.**: task-specific metrics (F1/Accuracy), **Mem.**: memory in thousands of tokens. AR: Accurate Retrieval, TTL: Time Time Learning, LRU: Long Range Understanding. Best performance values are shown in **bold**.

---

**The system prompt used to answer questions**

You are a reasoning assistant with access to structured memory. Use the memories below to provide accurate, relevant, and comprehensive responses to user queries.

MEMORY STRUCTURE:

- Core Memory: Fundamental facts about the user (preferences, roles, goals, etc.)
- Semantic Memory: General knowledge, factual or conceptual information
- Episodic Memory: Specific personal experiences or events with time and context

CURRENT MEMORY STATE:

<core_memory>
{core_memory_content}
< \core_memory>

<episodic_memory>
{episodic_memory_content}
< \episodic_memory>

<semantic_memory>
{semantic_memory_content}
< \semantic_memory>

INSTRUCTIONS:
- Use the memories above to inform your responses
- If information is available in memory, reference it appropriately
- If memory is insufficient to answer a question, acknowledge this clearly
- Provide helpful and contextual responses based on the available memory
- Be concise but comprehensive in your answers

---

Figure 10: The prompt used to answer questions in Mem-$\alpha$.

