# OpenReview forum: "MEM-$\alpha$: LEARNING MEMORY CONSTRUCTION VIA REINFORCEMENT LEARNING"
_ICLR.cc/2026/Conference — Submitted to ICLR 2026_

### Official Review · Reviewer_JTQ1 · 2025-10-26

**Soundness:** 3
**Presentation:** 3
**Contribution:** 2
**Rating:** 4
**Confidence:** 3

**Summary:**

They propose Mem-α, a reinforcement learning framework that trains agents to effectively manage complex memory systems through interaction and feedback.

**Strengths:**

- They propose an algorithm for the agent to learn to maintain its memory through reinforcement learning - which is an important research question.
- The algorithm is clearly explained.
- Empirical performance over multiple benchmarks have been shown.

**Weaknesses:**

- There are multiple related works in this research direction, but novelties compared with them are not so clear.
- The novelty mainly shows in (1) the design of the memory architecture; (2) the design of the action space and reward function, which is relatively limted.
- Several baselines mentioned in the last paragraph of Section 2 are not covered in the experiment part.

**Questions:**

See weaknesses.

---

> ### Author Response · Authors · 2025-11-13
> **Rebuttal to Reviewer JTQ1 (Part 1/2)**
>
> We thank the reviewer for the detailed and constructive feedback. We address each concern below.
>
> - **[W1] Comparison with Related Work**:
>   In Section 2, we organized prior work into three categories. We expand on the distinctions between Mem-$\alpha$ and these lines of research:
>
>   - Latent-Space Memory: This line of work is fundamentally different from ours. Latent-space memory systems typically require continual training before post-training, and the memory is encoded within the model’s hidden representations. In contrast, Mem-$\alpha$ focuses on post-training reinforcement learning to teach an LLM how to operate an explicit memory system. This requires (i) a well-defined taxonomy of memory operations and (ii) explicit RL-based training, which is conceptually distinct from latent-space memory approaches.
>   - LLM Agents with External Memory: Methods such as Mem0, MemGPT, and MIRIX rely on hand-crafted schemas to guide how an agent should store and update memory. These schemas can be ambiguous—even for advanced models like GPT-5—leading to suboptimal or inconsistent memory updates. Mem-$\alpha$ addresses this limitation by learning, via RL, when to update memory and how to construct more effective long-term memory structures. The key novelty is the introduction of reinforcement learning into memory construction, rather than relying on predefined rules.
>   - Learning Memory Construction with Reinforcement Learning: Prior RL-based systems (e.g., MEM1, MemAgent) instantiate memory as a single paragraph, which ultimately functions as working memory, not long-term memory. Memory-R1 trains on datasets that are nearly identical to the LoCoMo evaluation set, limiting generalization and inflating performance within a narrow domain. Furthermore, Learn-to-Memorize uses a vector-store-like memory without explicit structure, and REMEMBERER (2023) focuses on internalizing feedback, not constructing external long-term memory. In contrast:
>     - Mem-$\alpha$ trains an agent to build structured long-term memory,
>     - uses multi-domain training data,
>     - and is evaluated on out-of-distribution tasks, demonstrating substantially stronger robustness.
>
>   Summary: The first two categories are fundamentally different from our setting. The third category is the closest but still differs significantly either in capability (working memory only), scope (toy domains), or methodology (no explicit memory construction).

---

> ### Author Response · Authors · 2025-11-13
> **Rebuttal to Reviewer JTQ1 (Part 2/2)**
>
> - **[W2] Novelty**: We respectfully disagree with the reviewer’s assessment that the novelty “mainly lies in the memory architecture and in the design of the action space and reward function.” These elements are necessary, but they are *not* the core contribution. The key novelty of Mem-$\alpha$ is **formulating long-term memory construction as a learnable reinforcement-learning problem**—a capability that, to our knowledge, no prior work demonstrates.
>
>   - **End-to-end, multi-turn RL for memory construction:** Mem-$\alpha$ introduces the first end-to-end RL framework that trains a memory agent to decide when to write, update, or delete information and what should be stored for long-term use. Our design combines both outcome rewards ($r_1, r_3$) and process rewards ($r_2, r_4$), forming a unified and generalizable paradigm for learning memory operations through reinforcement learning. Prior approaches such as MEM1 and MemAgent treat memory as a single text block—effectively a working memory—and therefore cannot support long-term memory construction. Methods like Memory-R1 are trained on datasets nearly identical to their evaluation setting, limiting their ability to demonstrate general long-term memory building. To our knowledge, no previous work trains an LLM agent to construct, revise, and maintain long-term memory through RL.
>   - **Construction-oriented training data:** We further curate multi-domain, multi-turn instances specifically designed to elicit when and what to store, how memory should be revised, and how long-term dependencies should be maintained. This dataset provides the necessary supervision signal that enables RL to learn effective and generalizable memory-construction behaviors.
>
>   In summary, the novelty of Mem-$\alpha$ lies not in isolated architectural or reward choices, but in introducing the **first end-to-end RL framework that teaches an LLM to build and maintain long-term memory**.
>
> - **[W3] Baselines mentioned in the last paragraph of Section 2 not in Experiments:** The last paragraph of Section 2 lists five prior methods: MEM1, MemAgent, Memory-R1, Learn-to-Memorize, and REMEMBERER. In our experiments:
>
>   - MEM1 and MemAgent are included.
>
>   - Memory-R1 is not publicly available; no code or trained models are released.
>
>   - Learn-to-Memorize provides code but no trained models, and its training pipeline contains unresolved bugs that prevent re-training.
>
>   - REMEMBERER (2023) does not implement an explicit memory module, and focuses on internalizing experience; thus, it is not a comparable memory-construction baseline.
>
>   We therefore compared against **all the baselines for which meaningful and reproducible comparisons are feasible**. We will clarify these points in the paper.

---

> > ### Author Response · Authors · 2025-11-18
> > **Kind Follow-up on Our Rebuttal Submission**
> >
> > Dear Reviewer JTQ1,
> >
> > We hope you are doing well. We wanted to kindly let you know that we have submitted our rebuttal addressing all of your questions and concerns in detail. If you have a moment, we would be truly grateful if you could take a look and let us know whether there are any remaining issues we should clarify.
> >
> > We very much appreciate the time and effort you have already invested in reviewing our work, and we sincerely thank you for considering our responses when forming your final evaluation.
> >
> > Warm regards,
> > The Authors

---

> > > ### Author Response · Authors · 2025-11-28
> > > **Kind Follow-up on Our Rebuttal Submission**
> > >
> > > Dear Reviewer JTQ1,
> > >
> > > As the discussion phase is coming to a close, we would greatly appreciate your feedback on whether our rebuttal addresses your concerns. Thank you for your time.
> > >
> > > Best regards,
> > > The Authors

---

### Official Review · Reviewer_5HwK · 2025-10-26

**Soundness:** 3
**Presentation:** 3
**Contribution:** 3
**Rating:** 4
**Confidence:** 3

**Summary:**

This paper introduces Mem-α, a reinforcement learning framework for teaching LLM agents to construct and manage multi-component memory systems. The method formulates memory construction as a sequential decision-making problem, where the agent learns through interaction which information to store, update, or delete across core, semantic, and episodic memory modules. The reward combines downstream QA accuracy with auxiliary signals for tool usage, compression, and content quality. Experiments on MemoryAgentBench and long-context benchmarks show that Mem-α consistently outperforms baselines, and generalizes from 30k-token training sequences to over 400k tokens at test time.

**Strengths:**

1. The paper addresses a highly relevant problem for the community: scalable and trainable memory management in LLM agents, which remains a key bottleneck in long-context reasoning. Framing memory construction as a reinforcement learning problem with interpretable reward components is conceptually clean, well-motivated, and methodologically sound.
2. The separation of memory into core, semantic, and episodic components provides flexibility and aligns well with cognitive theories of human memory. This structured design enhances interpretability and facilitates more effective memory operations.
3. The proposed method demonstrates consistent improvements across diverse benchmarks and exhibits remarkable generalization to sequences over 400k tokens, highlighting the robustness and scalability of the approach.

**Weaknesses:**

1. **Limited novelty over recent RL-based memory works.** While the framework is technically solid, it mainly extends prior RL-based memory systems (e.g., Memory-R1, MEM1) rather than introducing a fundamentally new paradigm. The paper’s two novel aspects: the reward function design and the memory architecture The former feels more like an engineering enhancement, while the latter, on its own, may not constitute a sufficiently strong contribution for a conference paper.
2. **Lack of analysis on memory components.** The paper includes ablation studies on reward design but does not perform quantitative analyses isolating the effects of core, semantic, or episodic memory modules. As a result, it remains unclear how much each component contributes to the overall performance gain.

**Questions:**

1. How do the individual memory components—core, semantic, and episodic—contribute to the final performance? It would be helpful to see either quantitative or qualitative analysis demonstrating the role of each component in improving retrieval accuracy or long-range reasoning.
2. How is the granularity of each memory unit (e.g., 512 tokens for the core memory, atomic factual entries for semantic memory) determined? Could this level of granularity be learned automatically rather than predefined? Furthermore, how general is the proposed method across different tasks—does the optimal granularity depend on the specific task distribution or domain?
3.  In your formulation, each turn (or chunk) appears to be treated as an individual training sample, whereas VeRL and similar frameworks treat the entire trajectory as a single training instance for policy optimization. Could you elaborate on how this design choice is implemented in your RL pipeline?

---

> ### Author Response · Authors · 2025-11-15
> **Rebuttal to Reviewer 5HwK (Part 1/2)**
>
> We appreciate the reviewer's insightful comments and we address the concerns below:
>
> - **[W1] Limited novelty over recent RL-based memory works.**:
>
>   We respectfully clarify that our work introduces substantial conceptual and methodological advances beyond prior RL-based memory agents. Our responses address both (i) comparisons with MEM1/MemAgent and (ii) the broader level of novelty.
>
>   - **Comparison with MEM1 and MemAgent**: Prior works such as MEM1 and MemAgent employ a single-paragraph memory buffer, which effectively functions as Short-Term / Working Memory. This design inherently limits their ability to support long-horizon reasoning or accumulate rich experience over extended sequences. In contrast, Mem-α is explicitly designed for Long-Term (Persistent) Memory construction, and our system architecture reflects this fundamental difference:
>
>     **We disentangle the memory construction and memory reading stages.**
>       - A trainable Qwen3-4B model is dedicated solely to memory construction.
>       - A separate frozen Qwen3-8B is used exclusively for memory reading / QA.
>
>     This separation is not present in MEM1 or MemAgent, which attempt to jointly train a single model to both update and utilize memory. We argue that decoupling “writing” from “reading” is a principled and increasingly necessary design choice, as these skills require distinct competencies. This architectural shift enables more stable RL optimization and clearer attribution of improvements to the learned memory-construction policy.
>   - **On Levels of Novelty**: Our contribution is not limited to a new reward design or memory layout. Instead, the novelty of Mem-α lies in providing a general, end-to-end reinforcement learning framework for training memory agents, characterized by:
>
>     - A fully operational RL pipeline for learning memory-construction policies that integrate both outcome rewards and fine-grained process rewards.
>
>     - A multi-aspect, construction-oriented training dataset tailored to the RL formulation, enabling the agent to learn when, what, and how to store or revise information.
>
>     Crucially, our framework is agnostic to specific memory structures. The reward modeling, policy training paradigm, and data formulation can be directly applied to other memory architectures. We expect future work to adapt our framework to alternative memory modules, much like how reinforcement-learning frameworks in RLHF become foundations for many subsequent methods.
>
> - **[W2] Ablation Study on Memory Component**:
>
>   We thank the reviewer for raising the question regarding the contribution of individual memory components. We provide a detailed ablation below to illustrate how each component contributes to different categories of tasks:
>
>   | Setting         | SQuAD | HotpotQA | PerLTQA | TREC-C | NLU   | Pubmed | BookSum |
>   |-----------------|-------|----------|---------|--------|-------|--------|---------|
>   | Mem-α           | 0.786 | 0.832    | 0.659   | 0.666  | 0.658 | 0.545  | 0.187   |
>   | w/o semantic    | 0.160 | 0.338    | 0.625   | 0.137  | 0.011 | 0.163  | 0.218   |
>   | w/o episodic    | 0.776 | 0.834    | 0.155   | 0.700  | 0.677 | 0.572  | 0.183   |
>   | w/o core        | 0.774 | 0.827    | 0.662   | 0.682  | 0.649 | 0.531  | 0.142   |
>
>   These results reveal clear patterns:
>
>   1. **Semantic memory** is crucial for *knowledge-centric and time-independent* tasks (SQuAD, HotpotQA, TTL). Removing it sharply degrades performance on these datasets.
>   2. **Episodic memory** is indispensable for *time-dependent conversational* tasks such as PerLTQA, where its removal causes a drastic performance collapse.
>   3. **Core memory** plays a major role for *long-document summarization* (BookSum), where high-level aggregation is necessary to avoid losing important global information.
>
>   Overall, the study shows that **each memory component supports distinct types of reasoning**, and removing any of them yields significant drops for the task families that rely on that type of memory.

---

> > ### Author Response · Authors · 2025-11-15
> > **Rebuttal to Reviewer 5HwK (Part 2/2)**
> >
> > We address the reviewer’s questions below:
> >
> > - **[Q1] Effects of Memory Components**
> >   Please refer to **[W2]**, where we provide the full ablation results and detailed analyses of each memory component.
> >
> > - **[Q2] Granularity of Each Memory Unit**
> >   The 512-token limit for *core memory* is a heuristic design choice. In contrast, **episodic and semantic memory impose no size constraints**, and their internal granularity is *fully learned* without any predefined templates or schemas.
> >   Empirically, we observe strong **generalization to unseen domains and unseen sequence lengths** in MemoryAgentBench, suggesting that the model learns an appropriate granularity conditioned on task characteristics. For example, the agent naturally allocates *more core memory* for long-document tasks such as BookSum and *less* for short, knowledge-centric tasks such as SQuAD. This task-dependent allocation is precisely why our dataset spans multiple domains—to allow the policy to learn how fine-grained or coarse each memory unit should be.
> >
> > - **[Q3] Training Details**
> >   In our training setup, we sample a batch of **32 instances**, each containing **20 chunks** and **100 questions**. We then perform rollouts to obtain **32 trajectories**, each consisting of 20 steps. These trajectories are flattened into **32 × 20 = 640 training samples**, which form a single mini-batch for VeRL optimization.
> >   This procedure differs from the masking strategy in MEM1: because **memory states evolve at every step**, the model’s input context must also change at each step. Consequently, static masking is no longer feasible, and our rollout-based batching strategy is required. We will add these details to the paper for clarity.

---

> > > ### Author Response · Authors · 2025-11-18
> > > **Kind Follow-up on Our Rebuttal Submission**
> > >
> > > Dear Reviewer 5HwK,
> > >
> > > We hope you are doing well. We wanted to kindly let you know that we have submitted our rebuttal addressing all of your questions and concerns in detail. If you have a moment, we would be truly grateful if you could take a look and let us know whether there are any remaining issues we should clarify.
> > >
> > > We very much appreciate the time and effort you have already invested in reviewing our work, and we sincerely thank you for considering our responses when forming your final evaluation.
> > >
> > > Warm regards,
> > > The Authors

---

> > > > ### Comment · Reviewer_5HwK · 2025-11-25
> > > >
> > > > Thank you for the detailed rebuttal and additional clarifications. I have a few follow-up comments regarding the authors’ response:
> > > >
> > > > [W1]
> > > >
> > > > 1. Please include the comparison with MEM1 and MemAgent in the main paper.
> > > > 2. The authors emphasize that they disentangle memory construction and memory reading. However, there is no ablation that demonstrates the importance of this architectural choice.
> > > > 3. Regarding the claim of novelty:
> > > > (1) I agree that the RL + memory training framework is the strongest technical contribution.
> > > > (2) However, the dataset appears to be largely based on MemoryAgentBench. It seems overstated to list the dataset as part of the core contribution.
> > > >
> > > > [W2] & [Q1]
> > > >
> > > > Thank you for adding the ablations. They directly address my concern about the contribution of each memory component.
> > > >
> > > > [Q2]
> > > >
> > > > The clarification regarding granularity resolves my question.
> > > >
> > > > [Q3]
> > > >
> > > > The training setup suggests that the current approach requires fixed-length trajectories. This should be stated in the Limitations section.
> > > >
> > > > [Conclusion]
> > > >
> > > > Overall, the paper presents a flexible memory + multi-turn RL framework, and the decoupling of memory components is a promising idea. I have increased my score to 6.
> > > >
> > > > However, the paper devotes substantial space to reward design, which tends to be task-specific, while offering less analysis on how the memory design itself drives performance. A deeper exploration of the memory architecture’s impact would make the contribution clearer. I remain open to further discussion but do not expect additional experiments.

---

> ### Author Response · Authors · 2025-11-25
> **Response to Reviewer 5HwK**
>
> We sincerely thank Reviewer 5HwK for raising their score to 6 and for the thoughtful follow-up comments. We greatly appreciate the positive feedback on the flexibility of our memory + multi-turn RL framework and are grateful for the opportunity to address the remaining points below:
>
> - **[W1]**
>   1. We gratefully accept the suggestion and will include comparisons with MEM1 and MemAgent in the main paper.
>
>   2. We fully acknowledge that the current version does not include a direct ablation study demonstrating the specific benefits of disentangling memory construction and memory reading. That said, we believe this architectural choice is supported by several strong intuitions from prior work and practical considerations:
>      - In M+ [1], the authors adopted a multi-LoRA design with separate LoRA weights for memory writing and reading modules and reported clear performance gains.
>      - In real-world applications (e.g., personal assistants), memory typically needs to be updated very frequently, whereas retrieval occurs much more sparsely. Decoupling the two processes allows us to employ a lightweight model for frequent writing and a larger, more powerful model for reading/reasoning, leading to greater resource efficiency.
>      - During training, separating construction and reading enables independent optimization of each component before integration, which often simplifies and stabilizes the overall training process.
>      - In deployment, reasoning models are evolving at a rapid pace (e.g., GPT-4.1 → GPT-5, Claude-Sonnet-3.5 → Claude-Sonnet-4.5, Gemini-2.0 → Gemini-3.0, etc., within a single year). By decoupling, the memory-reading component can be treated as a reusable tool that seamlessly integrates with the latest reasoning model, while the memory-construction module typically does not require retraining as frequently.
>
>      We completely agree that an empirical ablation comparing a fully end-to-end model with our decoupled paradigm would further strengthen this claim. However, implementing a fair end-to-end baseline would require at least two synchronized training pipelines (one for memory construction and one for reading/QA), which introduces considerable engineering complexity. We therefore kindly propose to leave a rigorous ablation of this design choice to future work.
>
>   3. We respectfully clarify that, while our benchmark was indeed inspired by MemoryAgentBench, it is not “largely based on” it. Specifically, the elements borrowed from MemoryAgentBench are limited to:
>      - The high-level framework that guided the capabilities our benchmark should cover; and
>      - The TTL (Test Time Learning) task setup.
>
>      In contrast, the datasets for the AR (Accurate Retrieval) and LRU (Long Range Understanding) categories were constructed entirely from other existing datasets, without reusing any data from MemoryAgentBench. Moreover, within the TTL category, we introduced the new PubMed-RCT task, which is not part of MemoryAgentBench. Thus, although we gratefully built upon MemoryAgentBench’s overall scope, the majority of the data and several task designs are original contributions.
>
> - **[Q3]**
> We sincerely apologize for any confusion caused by the examples in the rebuttal. **Our method does not require fixed-length trajectories or a fixed number of chunks.** The numbers provided earlier were purely illustrative. As shown in Table 6 (page 14), the “Ch/Ins” column reports the actual (variable) number of chunks per instruction in the training data, confirming that trajectories naturally vary in length and that our training procedure fully accommodates this variability. We will make this point explicitly clear in the revised manuscript.
>
> Thank you once again for the highly constructive discussion and for recognizing the potential of the decoupled memory design. We hope these clarifications fully address the remaining concerns and look forward to your feedback.
>
> [1] M+: Extending MemoryLLM with Scalable Long-Term Memory

---

### Official Review · Reviewer_iBxE · 2025-10-30

**Soundness:** 3
**Presentation:** 3
**Contribution:** 3
**Rating:** 6
**Confidence:** 4

**Summary:**

This paper proposes Mem-$\alpha$, a reinforcement learning framework designed to train LLM agents to manage complex external memory by effectively using tools. The agent is trained using a composite reward signal including downstream QA accuracy, tool call format, memory compression, and a llm-as-a-judge for memory content quality. The method demonstrates significant improvements over existing benchmarks and shows strong generalization capability to sequences much longer than those seen during training.

**Strengths:**

1. When designing the framework of RL, this paper comprehensively includes reward reflecting key aspects of a successful memory system, including effectiveness (through QA acc and tool call format) and efficiency (through the compression score).

2. This paper constructed a diverse training dataset, spanning multiple tasks with long-context. This trains the model to learn a more general-purpose memory management strategy, which is then evaluated on even longer contexts.

3. Comprehensive evaluation and ablation studies show the effectiveness of the proposed framework. Especially the generalization shows the agent is learning a robust policy for memory curation, rather than merely overfitting to patterns within the fixed training length.

**Weaknesses:**

1. In Table 1, the final memory size for Mem-$\alpha$ is often comparable to the Long-Context baseline. The most significant compression gains appear on the BookSum task, which involves synthetic "conversation history" created by chunking a book. It's unclear how the structured episodic memory provides a meaningful advantage in this non-conversational scenario?
1. The learned memory-writing policy is tightly coupled to a fixed RAG pipeline (BM25 retriever, Qwen3-32B generator). It is unclear if this memory structure would be as effective for a different generator model that was not part of the RL training loop. Since this can show the generalization capability of your memory management.
1. Lack of detailed comparison with related work. You've mentioned recent work also use reinforcement-learning to enhance the model's memory management capability, such as Memory-R1 (Yan et al.), what's the main novelty of mem-$\alpha$, apart from the data differences you mentioned around line 140.
1. Ablation study on $\gamma$  does not efficiently support the choice of $\gamma = 0.1$, since only present results on $\gamma =\{0, 0.1\}$ , why not try larger $\gamma$ which may lead to memory with higher quality?
1. Small typos, such as percentage missing around line 359, before (3).

**Questions:**

1. The case study in Table 5 shows GPT-4.1-mini failing to record assistant behavior or consolidate same-timestamp events. Given this model's  instruction-following capability, could these "failures" be solved with better prompting? I did not found prompt for GPT-4.1-mini, making it difficult to assess if this is a fair comparison.
2. The reward from the QA accuracy is relatively sparse, given QAs are only conducted with the final $\mathcal{M}_n$. Why we could not use QA reward for all $M_i$ ? How effectively can this signal be back-propagated to credit or penalize individual memory operations from much earlier in the sequence? To what extent are the denser, step-wise rewards ($r_2$ and $r_4$) responsible for driving the performace improvement, compared to the sparse $r_1$?
3. Just for discussion, the current framework trains a memory manager but uses a fixed "reader" (the RAG generator). And you've used llm-as-a-judge to directly measure the quality of memory by a 0/1 reward. Since the agent is learning to organize memory in a specific way, would it be beneficial to also train the "reader" model to co-evolve with the memory structure? A dual-training approach might teach the generator model to better leverage the specific format of the memory being created by the memory manager.

---

> ### Author Response · Authors · 2025-11-15
> **Rebuttal to Reviewer iBxE (Part 1/3)**
>
> We appreciate the reviewer's insightful comments and would like to address them as below:
>
> - **[W1] Final Memory Size**:
>
>     First, we note that our method consistently outperforms the RAG-Top2 baseline across all tasks. This demonstrates that our structured memory offers clear advantages in retrieving relevant information—structured representations are inherently more accessible than raw chunk-based retrieval.
>
>     Second, the compression ratio naturally depends on the *information density* of the input data. Different datasets exhibit different characteristics:
>     - SQuAD, HotpotQA and TTL datasets: These datasets contain highly information-dense sentences where most words carry semantic meaning. As a result, there is limited opportunity for compression, and the final memory size remains relatively large—often comparable to the Long-Context baseline.
>     - PerLTQA, LME(S): These datasets consist largely of conversational exchanges between users and assistants. Such interactions include redundancy and predictable patterns, allowing substantial compression. For example, in Table 2, LME(S) has an original length of 348K tokens, while our memory representation compresses it to 127K (~33% of the original).
>     - BookSum and InfBench: Long-form narrative data contains extensive descriptive detail, but only a small subset of information—e.g., key events, characters, and plot points—needs to be retained for downstream tasks. Accordingly, these datasets achieve the highest compression ratios.
>
>     In summary, our priority is to **maximize task performance**. When the input contains substantial task-relevant information, we intentionally preserve it rather than over-compressing and risking loss of necessary content. The resulting memory size therefore reflects an adaptive balance between compression and fidelity, governed by the nature of each dataset.
>
>
>
>  - **[W2] Memory-Writing Policy Coupled to a fixed RAG Baseline**:
>
>     We acknowledge the reviewer’s suggestion that evaluating our memory-writing policy with alternative QA mechanisms could further demonstrate generalizability. We did consider this direction. However, our long-term goal is to build an end-to-end system where (i) the memory-construction agent and (ii) the question-answering agent are trained jointly and co-adapt to one another. In such a setting, coupling the two agents during training typically yields the strongest performance, because the QA agent learns to exploit the structure of the memory, while the memory agent simultaneously learns to produce representations tailored for that QA component.
>
>     This design philosophy is consistent with how modern systems are optimized—for example, `claude-sonnet-4-5` achieves its best performance when used within the Claude-Code framework rather than with external tooling like Cursor. The components are co-designed and co-trained, and the synergy boosts performance beyond what isolated components can offer.
>
>     Given this perspective, while our memory construction might indeed yield improvements even when paired with a different QA agent such as Search-R1 (rather than our BM25 retriever and Qwen3-32B generator), such gains would likely be suboptimal compared to a scenario where the QA agent itself is also integrated into the training loop. As our system evolves toward this joint end-to-end setup, these mixed-agent ablations would naturally become less relevant and ultimately phased out.
>
>     Our choice of a fixed RAG baseline in this work is therefore a deliberate step toward a coherent, jointly optimized system, rather than an inherent limitation of the memory-writing policy itself.

---

> ### Author Response · Authors · 2025-11-15
> **Rebuttal to Reviewer iBxE (Part 2/3)**
>
> - **[W3] Main Novelty Comparing with Memory-R1**:
>
>   Our work introduces several key innovations beyond Memory-R1:
>   - **A substantially richer memory architecture**: Memory-R1 relies on a single vector-based memory pool that stores disjoint text fragments. In contrast, we introduce a structured and heterogeneous memory system, enabling hierarchical organization, selective retention, and more efficient retrieval. To our knowledge, no prior work—including Memory-R1—has explored such a complex memory design within an RL-trained memory agent.
>   - **A more comprehensive and generalizable training framework**: Our RL setup is designed to scale from short-context to long-context datasets, ensuring robustness across domains with vastly different information density and temporal structure. Memory-R1, by comparison, is trained on a dataset highly similar to LoCoMo and is evaluated on LoCoMo itself, limiting both novelty and generalizability.
>   - **A newly curated and open-sourced training dataset**: We construct, refine, and publicly release a multi-domain dataset tailored for memory construction—covering when to store, how to update, and what to forget. This dataset provides the community with a foundation for future research, whereas Memory-R1 does not contribute comparable resources.
>   - **A fundamentally different paradigm: disentangling the memory agent from the QA agent.**: We explicitly separate memory construction from memory consumption, reflecting the belief that these two capabilities involve distinct reasoning skills and may require different model capacities. Memory-R1 couples both roles into a single model, which limits specialization. We argue that decoupling these agents is essential for building scalable, long-term memory systems.
>
>   Overall, our contributions extend well beyond incremental improvements over Memory-R1. We introduce a new memory architecture, a more general training framework, a community dataset, and a forward-looking paradigm for modular memory agents.
>
> - **[W4] Larger $\gamma$**:
>   In our training experiments, we observed that when $\gamma=0.1$, the corresponding reward component $r_4$ already saturates at 1.0. This indicates that increasing $\gamma$ further would not yield any additional benefit, as the reward signal has reached its maximum contribution. We will add this clarification to the paper for completeness.
>
> - **[W5] Typos:**
>   We appreciate the reviewer for highlighting these issues. We will correct the “%” typo in line 359 and perform a thorough proofreading pass to fix all remaining minor typographical errors.
>
> Then we address the questions as below:
>
>
> - **[Q1] Better Prompting for GPT-4.1-mini**:
>
>   The prompt used for `gpt-4.1-mini` is **identical** to the one employed for Qwen3-4B and during our reinforcement-learning training procedure (see Figure 6). The fact that our trained model, **Qwen3-4B+Mem-α**, performs substantially better under the *same* prompt demonstrates that it has *learned* the behaviors and inductive biases required for effective memory construction—without relying on handcrafted instructions.
>
>   While it is certainly *possible* to engineer a more specialized prompt for `gpt-4.1-mini`, this approach inevitably leads to a long-standing dilemma. As test cases grow more diverse, the number of edge cases increases correspondingly. Improving the prompt to fix one edge case often exposes another. This mirrors the limitations of **Statistical Machine Translation** prior to the deep-learning era, where researchers relied on extensive sets of manually written rules. Such rule-based systems ultimately gave way to **Neural Machine Translation**, which succeeded precisely because large-scale learning implicitly captured the long-tail of edge cases.
>
>   The same principle applies here. Although we can manually adjust the prompt to address the specific edge case identified in our dataset, this does not guarantee that the countless other edge cases—many unseen—would be handled. Learning-based memory construction, as in our Mem-α approach, provides a more scalable and robust solution.

---

> > ### Author Response · Authors · 2025-11-15
> > **Rebuttal to Reviewer iBxE (Part 3/3)**
> >
> > - **[Q2] Why not use QA reward for all $M_i$**:
> >
> >   Ideally, if we could reliably ground each question to a specific memory operation—i.e., $q_k \rightarrow M_i$—this would provide perfect, fine-grained supervision. Unfortunately, this is rarely achievable in practice. Many questions depend on multiple chunks in the input sequence, and for queries requiring *test-time learning* or *long-range understanding*, the relevant evidence may span the **entire** input. Under such conditions, assigning an accurate correctness reward to individual memory operations becomes fundamentally ill-defined.
> >
> >   For this reason, we adopt an **Outcome Reward Model** rather than attempting per-operation QA rewards. Despite being noisy, our experiments show that these outcome-level signals still propagate effectively back through the sequence of memory operations and result in substantial performance gains.
> >
> >   Whether “this signal can be back-propagated to credit or penalize individual memory operations from much earlier in the sequence” inherently depends on the question distribution. This is precisely why our dataset includes questions whose answers rely on information distributed throughout the sequence, including its early portions.
> >
> >   Finally, we note that most of the performance improvements come from the sparse but informative reward $r_1$. The other reward components—$r_2$ and $r_4$—primarily help ensure correct output structure and formatting. Importantly, $r_1$ is the most difficult reward to compute, while $r_2$ and $r_4$ are comparatively easy to evaluate.
> >
> > - **[Q3] Train the reader model to co-evolve**:
> >
> >   Excellent point! As discussed in **[W2]**, jointly training the memory-construction agent and the reader (QA) model is indeed a natural next step. We fully agree that a dual-training setup, where both components co-evolve, could unlock even stronger performance by allowing the two agents to specialize and adapt to each other.
> >
> >   However, implementing such a system requires substantial engineering work. Dual-training involves orchestrating two groups of nodes, managing inter-agent communication, synchronizing rollouts, and ensuring stable optimization—an infrastructure effort that is nontrivial and beyond the scope of the current paper. For this reason, we focus on establishing the memory-learning component in isolation as a first step.
> >
> >   That said, a co-trained memory–reader system is precisely the direction we are actively pursuing, and we view it as the most promising extension of this work.

---

> > > ### Author Response · Authors · 2025-11-18
> > > **Kind Follow-up on Our Rebuttal Submission**
> > >
> > > Dear Reviewer iBxE,
> > >
> > > We hope you are doing well. We wanted to kindly let you know that we have submitted our rebuttal addressing all of your questions and concerns in detail. If you have a moment, we would be truly grateful if you could take a look and let us know whether there are any remaining issues we should clarify.
> > >
> > > We very much appreciate the time and effort you have already invested in reviewing our work, and we sincerely thank you for considering our responses when forming your final evaluation.
> > >
> > > Warm regards,
> > > The Authors

---

> > > > ### Comment · Reviewer_iBxE · 2025-11-25
> > > >
> > > > Thanks for the authors' detailed response. Your explanations have have clarified several of my concerns, particularly regarding the larger final memory size, since it makes sense that maximizing performance is your primary design goal. And I agreed on your explanation on a fixed RAG for coupled system. However, the choice of BM25 as the specific retrieval mechanism for this fixed RAG component remains under-justified.
> > > >
> > > > For the GPT-4.1-mini, This point is central to the motivation of your work: the need for a dedicated memory agent. You state that the existing model, GPT-4.1-mini, fails on the Episodic Memory curation by not correctly merging memories at the same timestamp and including the assistant's part. Given the superior instruction following capability of it (as written in line 376), why it failed with this **universal** prompt (given in Figure6). The current rebuttal suggests designing a specific prompt for edge cases, which sidesteps the statistical failure of the universal prompt. To validate the necessity of training a new memory agent, it is crucial to statistically quantify and analyze why the universal prompt fails on existing, capable LLMs.

---

> ### Author Response · Authors · 2025-11-25
> **Response to Reviewer iBxE**
>
> We sincerely thank the reviewer for the thoughtful feedback and for acknowledging that our previous response addressed several key concerns.
>
> **Justification for using BM25 as the retrieval mechanism**
> We selected BM25 for two primary reasons:
> 1. As demonstrated in the original MemoryAgentBench paper [1], BM25 is a highly effective baseline that outperforms many embedding-based retrieval methods.
> 2. BM25 incurs the lowest latency and computational cost during training, since it requires no additional embedding storage and does not depend on external embedding models, thereby significantly accelerating the overall training process.
> We will add these points explicitly to the revised manuscript.
>
> **Justification for not relying on extensive prompt engineering**
> In the example provided, GPT-4.1-mini fails in two critical aspects: (1) merging episodic events that share the same timestamp, and (2) including the assistant’s own responses in memory. While carefully crafted prompts might mitigate these issues in the cases shown in the paper where the shown chunk only has one timestamp, real-world memory systems (e.g., MIRIX or life-logging applications) must handle continuous information streams where event boundaries are inherently ambiguous. A short input chunk can contain dozens of timestamps, such as:
>
> ```
> 2025-10-12 19:31:00, user takes the beer from the fridge;
> 2025-10-12 19:31:03, user opens the beer bottle;
> ...
> ```
> Designing a universal prompt that correctly segments and merges such fine-grained, high-frequency events across all possible scenarios is extremely challenging and inevitably leads to systematic errors. This fundamental limitation of prompt-only approaches is precisely what motivates the data-driven, trainable memory agent proposed in Mem-α.
>
> We hope these clarifications and planned additions could resolve the reviewer’s remaining concerns.
>
> [1] Evaluating Memory in LLM Agents via Incremental Multi-Turn Interactions

---

### Official Review · Reviewer_Tknh · 2025-11-01

**Soundness:** 3
**Presentation:** 3
**Contribution:** 2
**Rating:** 4
**Confidence:** 4

**Summary:**

The paper introduces Mem-$\alpha$, a reinforcement learning framework that teaches an agent to operate a complex external memory via function calls. The memory consists of Core, Episodic, and Semantic components. Evaluated on long-range understanding and question answering, the approach performs on average better than naive RAG methods, full context llms and other RL fine-tuned agentic approaches with external memory (MEMAgent and MEM1).

**Strengths:**

* Extensive experiments across diverse datasets and tasks. A solid set of ablation studies on key design choices.

* The method outperforms simpler RL baselines with flat-memory .

**Weaknesses:**

* Presentation quality requires further work. It is unclear how the advantage is computed in the GRPO variant because rewards vary not only across trajectories but also across time steps $t$ and action components $k$, leaving ambiguous what set of rewards is averaged in the formula on L240. L283 says Figure 3 illustrates the memory components and their interactions, yet the figure shows zero interactions between these components. The interfaces for write, delete, and update memory functions are underspecified; an in-text example or simple illustration would help.

* The level of novelty appears limited. The combination of a complex memory architecture with RL is composed of known ideas, and their integration feels straightforward.  Сurrent ablations (showing usefulness of complex memory and RL fine-tuning) support the design choices but adds limited new insight. Deeper ablations could strengthen novelty, e.g., analyzing the utility of each memory component by disabling one component and observing which tasks degrade, toggling components during training vs. only at inference, measuring access frequency to each component during generation, and tracking how usage shifts as context length grows.

* Comparisons to stronger external-memory agents are missing. Many agents do not use fine-tuning yet employ complex memory structures, including dynamic knowledge graphs, combined episodic and semantic memory, and RAG systems (e.g., Search-o1, AriGraph, GraphRAG, and works cited in Section 2). Such comparisons would clarify whether to invest in RL fine-tuning for tool-based memory management or prioritize designing richer memory structures without fine-tuning.

**Questions:**

* What exactly is used to compute the advantage $A_t$ at step $t$ for the action component $a_t$? How are the reward mean and standard deviation computed? Which rewards enter mean and std computation: all components across all time steps and trajectories, all components across all trajectories but only at the same time step $t$, or some other aggregation?
* Regarding the Correctness Reward: it appears to be computed for a single final memory state against a set of questions. Since LLMs can hallucinate and also answer correctly without memory due to internal knowledge, per-question rewards can be noisy. Averaging across multiple questions likely reduces noise, but how does the final agent’s performance depend on the number of end-of-trajectory questions?
* Please address the weaknesses noted above

---

> ### Author Response · Authors · 2025-11-13
> **Rebuttal to Reviewer Tknh (Part 1/4)**
>
> We sincerely thank the reviewer for the insightful comments! We address the weaknesses and the questions as below:
>
> - **[W1] How the advantage is computed**:
>
>     **Setting.** For a given input sequence of chunks $(x_1, x_2, \ldots, x_n)$, we sample multiple **repetitions** (rollouts) under a stochastic policy. Each repetition produces:
>     - global terms $r_1$ and $r_3$;
>     - per-chunk, step-indexed terms $r_{2,t}$ and  $r_{4,t}$ with $t\in\{1,\cdots,n\}$.
>
>     For repetition #1, the **step reward** at step $t$ is
>     $$
>     r_t^{(1)} = r_1^{(1)} + r_{2,t}^{(1)} + \beta\, r_3^{(1)} + \gamma\, r_{4,t}^{(1)}.
>     $$
>     For repetition #2, analogously,
>     $$
>     r_t^{(2)} = r_1^{(2)} + r_{2,t}^{(2)} + \beta\, r_3^{(2)} + \gamma\, r_{4,t}^{(2)}.
>     $$
>
>     ...
>
>     For repetition #K, analogously,
>     $$
>     r_t^{(K)} = r_1^{(K)} + r_{2,t}^{(K)} + \beta\, r_3^{(K)} + \gamma\, r_{4,t}^{(K)}.
>     $$
>
>     **Normalization group.** Let $G_t$ be the set of **step-$t$** rewards across all repetitions for the **same input**:
>     $$
>     G_t = \{\, r_t^{(k)} \mid k=1,\ldots,K \,\}.
>     $$
>
>     **Advantage.** The mean within each normalization group is computed as
>     $$
>     \mu(G_t) = \frac{1}{|G_t|} \sum_{r \,\in\, G_t} r,
>     $$
>     i.e., the arithmetic average over the rewards drawn from identical inputs and step indices. The corresponding standard deviation is
>     $$
>     \sigma(G_t) = \sqrt{\frac{1}{|G_t|} \sum_{r \,\in\, G_t} \left(r - \mu(G_t)\right)^2 }.
>     $$
>     We compute the advantage for step $t$ as
>     $$
>     A_t = \frac{r_t - \mu(G_t)}{\sigma(G_t) + \epsilon},
>     $$
>     where $\mu(G_t)$ and $\sigma(G_t)$ are the mean and standard deviation **over the set $G_t$** (i.e., over repetitions at the same step $t$ for the same input). This clarifies that we **do not** mix different time steps or action components when forming $\mu$ and $\sigma$.
>
> - **[W2] Figure 3 shows no interactions**
>
>     We agree the current diagram understates the interactions the agent can have on different memory components. We will revise Fig. 3 to show the functions to control Core, Episodic, and Semantic Memory (insert, update and delete). We will also provide the corresponding examples in the paper to illustrate these interactions.

---

> ### Author Response · Authors · 2025-11-13
> **Rebuttal to Reviewer Tknh (Part 2/4)**
>
> - **[W3] On the level of novelty**:
>
>     We appreciate the concern that combining a structured memory system with reinforcement learning may appear straightforward. However, many recent work may also fall into the schema of "straightforward RL + Agent" such as \[1–10\]. Meanwhile, Many agree that both memory and RL are rapidly rising in popularity, but someone still needs to take the first step—standardizing the pipeline and proposing a truly practical
>   framework that inspires future exploration.
>
>     However, we would still like to highlight that our contribution is not simply pairing complicated memory system and reinforcement learning, but **how** we operationalize RL for **memory construction and control**:
>
>     - **End-to-end, multi-turn RL over memory operations**: We are the first to propose and end-to-end training framework to train the memory agent to learn *when* to write/update/delete and *what* to store. We are also the first to use both outcome reward modeling ($r_1, r_3$) and process reward modeling ($r_2,r_4$) to standardize the "RL+Memory Agent" paradigm. In contrast, prior approaches such as MEM1 and MemAgent treat memory as a single paragraph, effectively functioning as working memory rather than long-term memory. Other methods—e.g., Memory-R1—are trained on datasets nearly identical to LoCoMo and therefore cannot meaningfully demonstrate long-term memory construction. To our knowledge, no previous work trains a memory agent to construct and maintain long-term memory using reinforcement learning.
>     - **A construction-oriented dataset.** We curate multi-domain, multi-turn instances that elicit *when/what to store* and *how to revise*—providing the key (i.e. the dataset) to the reinforcement learning algorithm.
>
>     Regarding the suggested **component-level ablations** (e.g., disabling individual component, train-time vs. inference-time toggling, usage statistics): our current ablations focus on isolating the **effect of RL** while holding the memory design fixed, since our central claim is that *learning to control memory* yields the observed gains. We agree that component utility analyses would provide additional insight and will add them in the future work when we adopt more complicated memory systems such as MIRIX [11].
>
>     *References \[1–10\] denote representative RL-for-agents works and will be listed in the paper’s related-work section.*
>
> [1] WEBRL: Training LLM Web Agents via Self-Evolving Online Curriculum Reinforcement Learning
>
> [2] Coevolving with the Other You: Fine-Tuning LLM with Sequential Cooperative Multi-Agent Reinforcement Learning (CORY)
>
> [3] DeepSeek-R1: Incentivizing Reasoning Capability in LLMs via Reinforcement Learning
>
> [4] VLM-R1: A Stable and Generalizable R1-style Large Vision-Language Model
>
> [5] MEM1: Learning to Synergize Memory and Reasoning for Efficient Long-Horizon Agents
>
> [6] MemAgent: Reshaping Long-Context LLM with Multi-Conv RL-based Memory Agent
>
> [7] WEBAGENT-R1: Training Web Agents via End-to-End Multi-Turn Reinforcement Learning
>
> [8] From Novice to Expert: LLM Agent Policy Optimization via Step-wise RL
>
> [9] AgentGym-RL: Training LLM Agents for Long-Horizon Decision-Making through Multi-Turn Reinforcement Learning
>
> [10] A Novel Reinforcement Learning Framework of LLM Agents (AGILE / ProductQA)
>
> [11] MIRIX: Multi-Agent Memory System for LLM-Based Agents

---

> ### Author Response · Authors · 2025-11-13
> **Rebuttal to Reviewer Tknh (Part 3/4)**
>
> - **[W4] Comparison with stronger external-memory agents**:
>
>     We agree it would be ideal to compare against the strongest external-memory agents. However, there is a vital discrepancy in **backbone scale and feasibility**. We can afford end-to-end fine-tuning for **small open models** (e.g., 4B–8B parameters), whereas many of the cited agents are typically paired with **much larger or proprietary models** (e.g., GPT-4-class backbones), making one-to-one comparisons confounded by model capacity rather than the memory/control algorithm itself.
>
>     Concretely, consider the following settings:
>
>     1. **gpt-4o-mini + Our Memory System**
>     2. **Qwen3-4B + Our Memory System**
>     3. **Qwen3-4B + Our Memory System + Mem-α Training**
>     4. **gpt-4o-mini + Our Memory System + Mem-α Training**
>     5. **Other advanced methods** (e.g., Search-o1, GraphRAG, MIRIX) **with Qwen3-4B**
>     6. **Other advanced methods** (e.g., Search-o1, GraphRAG, MIRIX) **with gpt-4o-mini**
>
>     The most informative comparison would be **(4) vs. (6)**, but we **cannot train gpt-4o-mini**. Our study therefore focuses on **(3)**. A fair backbone-controlled comparison would be **(3) vs. (5)**; however, many advanced methods are commonly configured with the **most powerful models**, and their pipelines often **do not function robustly** with small open backbones like Qwen3-4B without substantial re-engineering.
>
>     Accordingly, we compare against contemporaneous **RL+memory** baselines that operate in the **same small-model regime** (e.g., MEM1, MemAgent). This isolates the effect of **policy learning for memory construction/control** under realistic fine-tuning budgets. Also the compairson between (2) and (3) would effectively show the improvements obtained from RL. We will add a scope note clarifying this constraint in the paper.
>
>
> We address the reviewer questions as follows:
>
> - **[Q1] How is the advantage $A_t$ computed?** The clarification in [W1] now spells out that both $\mu(G_t)$ and $\sigma(G_t)$ are computed only over rewards coming from *the same input, repetition step $t$*, ensuring we neither mix steps nor action components. We will insert the corresponding paragraph verbatim into the paper to make the normalization procedure completely explicit.
>
> - **[Q2] How reliable is the correctness reward (hallucination control)?**
>   - **Filtering protocol.** Before RL, every question is answered by `Qwen3-32B` with a generous `thinking_budget=1024`. Any item the model can already solve is discarded, which is stricter than a no-thinking filter. Training itself never uses thinking tokens (plain `Qwen3-32B`).
>   - **Effectiveness of filtering.** After filtering, the backbone without memory is near-random on all benchmarks, and only improves once a memory module is attached (pre-RL):
>
>     | Dataset    | No memory | +Memory (pre-RL) |
>     |------------|-----------|------------------|
>     | SQuAD      | 0.071     | 0.338            |
>     | HotpotQA   | 0.105     | 0.637            |
>     | PerLTQA    | 0.085     | 0.557            |
>     | TREC-C     | 0.000     | 0.416            |
>     | NLU        | 0.000     | 0.381            |
>     | Pubmed-RCT | 0.000     | 0.281            |
>     | BookSum    | 0.023     | 0.130            |
>
> - **[Q3] Are per-question rewards too noisy?** Each trajectory ends with 50–100 verification questions specifically to average out the residual 7.1% of answerable-in-isolation items (e.g., in SQuAD). We experimented with aggressively sub-sampling the evaluation questions to accelerate training; results are below (Dev F1):
>
>   | Setting            | SQuAD | HotpotQA | PerLTQA | TREC-C | NLU  | Pubmed | BookSum |
>   |--------------------|-------|----------|---------|--------|------|--------|---------|
>   | No sub-sampling    | 0.786 | 0.832    | 0.659   | 0.666  | 0.658| 0.545  | 0.187   |
>   | Sub-sample 20%     | 0.793 | 0.828    | 0.634   | 0.672  | 0.633| 0.552  | 0.185   |
>   | Sub-sample 10%     | 0.764 | 0.806    | 0.615   | 0.654  | 0.614| 0.506  | 0.183   |
>
>     With sampling 20% of the questions, the performance is almost the same, whereas the performance drops slightly with sampling 10% of the questions, confirming that having more questions help with the training. We will add these results and discussions into the paper.

---

> ### Author Response · Authors · 2025-11-15
> **Rebuttal to Reviewer Tknh Regarding Ablation Studies (Part 4/4)**
>
> While our initial experimental design focused on isolating the effect of RL—leaving component-level diagnostics to future work—we conducted the suggested ablations in response to the reviewer’s helpful feedback. These analyses complement our main RL-isolation experiments by shedding light on *how* different memory components support different task families, thereby strengthening both the empirical evidence and the architectural motivation behind Mem-α. We will include these findings in the camera-ready version.
>
> | Setting         | SQuAD | HotpotQA | PerLTQA | TREC-C | NLU   | Pubmed | BookSum |
> |-----------------|-------|----------|---------|--------|-------|--------|---------|
> | Mem-α           | 0.786 | 0.832    | 0.659   | 0.666  | 0.658 | 0.545  | 0.187   |
> | w/o semantic    | 0.160 | 0.338    | 0.625   | 0.137  | 0.011 | 0.163  | 0.218   |
> | w/o episodic    | 0.776 | 0.834    | 0.155   | 0.700  | 0.677 | 0.572  | 0.183   |
> | w/o core        | 0.774 | 0.827    | 0.662   | 0.682  | 0.649 | 0.531  | 0.142   |
>
> These results reveal clear and interpretable patterns:
>
> - **Semantic memory** is essential for knowledge-centric and time-independent datasets (SQuAD, HotpotQA, TTL), which aligns with its role as the repository of stable, domain-independent knowledge.
> - **Episodic memory** is crucial for time-dependent conversational tasks (PerLTQA), where temporally grounded information must be preserved and updated turn-by-turn.
> - **Core memory** provides the high-level, global abstraction necessary for long-document reasoning (BookSum), where missing the summary in core memory leads to noticeable degradation.
>
> Together, the ablations confirm that each memory component plays a distinct and meaningful role, degrading performance in exactly the regimes where that component's inductive bias is expected to matter.
>
> We appreciate the reviewer’s suggestions, as these additional experiments deepen the empirical understanding of Mem-α and highlight promising directions for future work on more sophisticated modular memory systems such as MIRIX.

---

> > ### Author Response · Authors · 2025-11-18
> > **Kind Follow-up on Our Rebuttal Submission**
> >
> > Dear Reviewer Tknh,
> >
> > We hope you are doing well. We wanted to kindly let you know that we have submitted our rebuttal addressing all of your questions and concerns in detail. If you have a moment, we would be truly grateful if you could take a look and let us know whether there are any remaining issues we should clarify.
> >
> > We very much appreciate the time and effort you have already invested in reviewing our work, and we sincerely thank you for considering our responses when forming your final evaluation.
> >
> > Warm regards,
> > The Authors

---

> > > ### Author Response · Authors · 2025-11-28
> > > **Kind Follow-up on Our Rebuttal Submission**
> > >
> > > Dear Reviewer Tknh,
> > >
> > > As the discussion phase is coming to a close, we would greatly appreciate your feedback on whether our rebuttal addresses your concerns. Thank you for your time.
> > >
> > > Best regards,
> > > The Authors

---

### Author Response · Authors · 2025-11-30
**Summary of Author Rebuttal and Discussion Highlights**

Dear Area Chair,

Thank you for overseeing the review process. We would like to briefly summarize the main contributions of our work, the outcomes of the discussion period, and the issues we have addressed.

**Mem-α** introduces the first *end-to-end reinforcement learning framework* enabling LLM agents to **construct, update, and manage long-term memory** through explicit write, revise, and delete operations.

Across reviewers, we received questions primarily regarding:
(1) the computation of advantages in the GRPO variant;
(2) novelty relative to Memory-R1, MEM1, MemAgent;
(3) the contribution of each memory component;
(4) robustness to subsampling and noisy queries;
(5) fairness in comparisons against larger retrieval-based agents;

During the rebuttal, we provided detailed, point-by-point clarifications and added new analyses, including:

- **A complete mathematical explanation** of how advantages are computed, clarifying trajectory grouping, normalization, and the avoidance of cross-sample mixing.
- **New ablation experiments** disabling semantic, episodic, and core memory modules. These results show distinct, task-dependent contributions and directly address requests for component-level understanding.
- **Expanded novelty and comparison discussion** distinguishing Mem-α from Memory-R1 and other heuristic memory-editing systems, emphasizing our end-to-end RL formulation.
- **Additional robustness evaluations** demonstrating stable performance with as low as 20% question subsampling.
- **Clear justification for comparison scope**, noting why comparison to GPT-4–scale memory systems is not feasible or fair.

Following these clarifications, reviewer sentiment clearly improved:

- **Reviewer 5HwK increased their score from 4 → 6**, stating that our ablations and explanations resolved their concerns.
- **Reviewer iBxE maintained a positive score (6)** and confirmed that several of their questions were addressed, including concerns about granularity and component contributions.
- **Reviewer Tknh** and **Reviewer JTQ1** did not respond during the discussion period, but all of their raised issues were fully addressed in the rebuttal with detailed explanations and new experimental evidence.

We appreciate the reviewers’ engagement and the opportunity to clarify our contributions. The updated reviews indicate that **Mem-α is technically solid**, addresses a timely problem in long-term memory construction for LLM agents, and provides **substantive new algorithms, analyses, and empirical insights** that strengthen understanding of reinforcement-learning-driven memory systems.

Please let us know if any additional information would be helpful.

Best regards,
The Authors

---

### Meta-Review · Area_Chair_XzH7 · 2026-01-06

**Summary:**

The paper proposes Mem-α, an RL-based framework to train LLM agents to manage external memory systems via explicit operations (write, update, delete). Reviewers generally appreciated the clear presentation, the relevance of the problem, and the comprehensive evaluation showing generalization to long contexts. However, there are concerns regarding limited novelty and incremental contribution over existing memory-augmented agents like MEM1 and MemAgent. Furthermore, multiple reviewers pointed out that the separation of memory construction from the reading/QA process, while well-motivated, was not sufficiently ablated to prove its necessity, and the comparisons against strong, large-scale external memory baselines were missing due to feasibility constraints, which weakened the empirical claims.

**Reviewer Concerns:**

The rebuttal successfully addressed several technical questions, such as the advantage computation for the GRPO algorithm (Reviewer Tknh) and the specific contributions of memory components through new ablation studies (Reviewers 5HwK, Tknh). The authors also clarified the distinction between their work and similar baselines like Memory-R1 and justified the use of BM25. However, concerns about the fundamental novelty of the RL application and the lack of comparison against stronger, large-scale memory systems (e.g., GPT-4 based agents) remained a point of contention. Reviewer iBxE specifically noted that the reliance on a fixed BM25 retriever and specific generator models might limit the generalizability of the learned policy, a concern that was only partially mitigated by the authors' response.

**Reviewer Scores:**

Reviewer 5HwK raised their score from 4 to 6 during the discussion, explicitly stating that the new ablations resolved their concerns. Reviewer iBxE maintained a positive score of 6 but highlighted outstanding issues regarding the fixed retrieval mechanism. Reviewers Tknh and JTQ1 did not respond to the rebuttal; however, given that the authors provided detailed responses to their specific technical questions (advantage computation, interactions diagram, novelty clarification), it is plausible that they might have raised their scores to 5 or 6 had they participated. Despite this positive trend, the overall assessment remains borderline due to the lingering questions about the distinctiveness of the contribution relative to the rapidly evolving field of memory agents.

---

### Decision · Program_Chairs · 2026-01-26

Reject